# Reminders of Japanese redress increase Asian American support for Black reparations

Michael W. Kraus [1,2,3] ✉ & A. Chyei Vinluan [1]

Informational interventions can shape policy attitudes, and in this study, we examined whether largely unknown information about past reparations payments toward one minoritized group would shape current policy judgments. In 1942, the U.S. government wrongfully relocated and imprisoned more than 120,000 Japanese Americans. In 1988, the government apologized and offered $20,000 USD in reparations payments. Japanese American redress is a recent, but not widely known, concrete example of communities who have successfully fought for reparative economic action. In two preregistered studies of online crowdsourced panels of Asian Americans ($N = 329$, $N = 500$), an intervention that raised awareness of this history of incarceration and redress increased support for reparations for Black Americans, relative to a control condition, and national polling data on support for reparations. Exploratory analyses revealed that the degree of learning about Japanese American redress in the intervention explained its impact on support for Black reparations. Future research should target representative samples to understand how education about past redress within one's own social group affects support for reparative economic justice for others.

[1] Yale University, School of Management, New Haven, CT, USA. [2] Yale University, Department of Psychology, New Haven, CT, USA. [3] Northwestern University, Department of Psychology, Evanston, IL, USA. ✉email: michael.kraus@northwestern.edu

In 1942, after the Japanese navy bombed the US military base in Pearl Harbor, the US government deemed Japanese Americans a threat to national security and then President Franklin D. Roosevelt signed executive order 9066, which forced relocation and imprisonment on more than 120,000 Japanese Americans[1]. Japanese American families were forcibly removed from their homes and relocated to prison camps. Though recounting of the Japanese American incarceration can be found in most history curricula, what is perhaps less well known is the struggle in the intervening decades for redress and reparation. This struggle resulted in the Civil Liberties Act of 1988 where the US government apologized for the incarceration of Japanese American families and offered a payment of $20,000 USD for harms caused by relocation and imprisonment[1]. In this research, we ask if knowledge of Japanese American redress will increase Asian American support for another movement for reparative economic justice: Reparations payments for Black Americans for chattel slavery and Jim Crow[2].

We build on prior research in the social sciences on Asian and Black intergroup relations[3,4], solidarity between minority groups[5–7], and on informational intervention strategies for policy change[8–10]. Specifically, we investigate whether a sample of Asian Americans exposed to information about this moment in Japanese American history—which we will refer to here as Japanese incarceration and redress—will, in turn, support reparations for Black Americans more than those in a control condition.

Understanding the history of relations between Asian and Black Americans is critical for a deeper understanding of movements for reparative justice. Though histories of racism differ considerably between these groups, Asian and Black American intergroup relations have been characterized by solidarity. In the US, Asian civil rights activists like Yuri Kochiyama and Thich Nhat Hanh worked in tandem with Black leaders like Malcolm X and Martin Luther King Jr. to fight for racial justice[11]. Contemporary groups including Asians for Black Lives carry out this tradition of solidarity[12].

Alongside this history of solidarity has been intergroup conflict. Scholars often discuss this conflict in terms of racial positioning, wherein two minoritized groups (e.g., Asian and Black Americans) are positioned against each other through rhetoric and policy in an attempt to drive discord between those groups[13]. For example, model minority stereotypes are one form of positioning, wherein Asian Americans are conceived of, through rhetoric and policy, as a so-called problem free minority group, and then this stereotype is used to legitimize and maintain unjust racial hierarchy by suggesting that non-Asian minority groups themselves are the problem[14,15]. When these model minority messages are internalized by Asian Americans, conflict can arise on policies for achieving racial justice. For instance, in a 2023 survey by Pew, 52% of Asian Americans disapproved of selective colleges considering race in admissions decisions (i.e., affirmative action)[16]. In contrast, 71% of Black Americans approved of the policy[16]. In this research, we attempt to better understand ways to promote solidarity and reduce ongoing potential conflict between Asian and Black Americans.

Payments have long been promised by the U.S. government to Black Americans for the grave injustice of chattel slavery and Jim Crow[2]. Despite these promises, and the documented negative economic effects of U.S. racism[17,18], recent polling data indicates that Americans are largely opposed to reparations for Black American descendants of slavery, with a 2021 Amherst poll finding that 64% of Americans, 47% of Asian Americans, and 90% of Republicans reported that they oppose the policy[19].

One route to reparations for Black Americans exists in the enactment of policy, and in this domain, public opinion is critical. Given their past solidarity with Black Americans on other issues and in activism for racial justice, Asian Americans may be particularly likely to support reparations for Black Americans under certain conditions. One way to promote support for reparations among Asian Americans is through an informational intervention that could remind this community of past redress impacting Japanese Americans following their relocation and incarceration. To the extent that Asian Americans are unaware of past redress for Japanese Americans, such an intervention could suggest to Asian Americans that redress payments should be extended to other marginalized groups.

Several lines of evidence indicate that an informational reminder of past redress for Japanese incarceration could increase support for Black reparations payments among Asian Americans. Prior research indicates that the effectiveness of teaching an accurate history on attitude change may depend on the message being new to the target audience in the first place. For instance, reminders of Black-White wealth inequality in the context of data on inequality in wealth, education funding, health, and housing increased accuracy in perceptions of the Black-White wealth gap that lasted for up to 18 months after the intervention[9]. This type of informational intervention may have worked because Black-White wealth inequality is a fact that Americans are largely unaware of, or unwilling to grapple with, as several studies attest[20,21].

Evidence indicates that there is a lack of knowledge of Japanese redress among Americans, and Asian Americans in particular, which indicates that the informational intervention may be effective: A YouGov 2014 poll found that a minority of Americans (37%) were in favor of Japanese reparations payments, a finding which reflects that many Americans may not know about the history of these payments already having occurred, or much Asian American US history in general[22]. This is compounded by the fact that many Japanese American people who were incarcerated were unwilling to talk about this period of their lives within their own communities[1]. Together, these data indicate that Americans in general, as well as Asian Americans more specifically, may be largely unaware of the successful movement toward Japanese redress, and as a result, may be less supportive of reparations for Black Americans than they otherwise would be if this information were known.

A second goal of our research is to better understand why an intervention that provides knowledge of Japanese redress might increase Asian American support for Black reparations. We contend that learning about a history of redress for one's own community will create conditions for moral hypocrisy that Asian Americans will seek to avoid by supporting similar redress for Black Americans.

Moral hypocrisy is derived from past social psychological research on cognitive dissonance[23]. In that work, having participants (a) learn about or experience a positive outcome for their own group, and then (b) receive reminders of a failure of another group to experience that same or similar outcome induces a discrepancy called moral hypocrisy. Moral hypocrisy is an uncomfortable, dissonant state, and thus, people seek to resolve the discrepancy through changes in attitudes that create consistency in treatment between groups[5,24]. In recent research, this paradigm has been applied to religious groups: Participants were less likely to collectively blame all Muslims for transgressions of individual Muslims when reminded that they do not engage in similar collective blame for Christians[5]. In follow-up research, a similar moral hypocrisy intervention reduced support for anti-Muslim policies[24].

We reason that providing information about the successful implementation of Japanese redress to a sample of Asian Americans and then asking them about unrealized redress for Black Americans due to chattel slavery and Jim Crow will engender a

**Table 1 Summary of demographic characteristics of Asian American samples from Study 1 and 2.**

| Variable | Study 1 | Study 2 |
|---|---|---|
| Sample size | $N = 329$ | $N = 500$ |
| Largest Asian subgroup | Chinese ($n = 91$) | Chinese ($n = 146$) |
| Largest US immigrant generation | First ($n = 166$) | Second ($n = 326$) |
| Women | $n = 163$ | $n = 186$ |
| Median income | $Mdn = \$60,001–\$80,000$ | $Mdn = \$80,001–\$100,000$ |
| Mean age | $M = 43.80$ ($SD = 18.21$) | $M = 31.98$ ($SD = 9.80$) |
| Median education | $Mdn = $ College degree | $Mdn = $ College degree |

desire to avoid hypocrisy. We expect participants to avoid this discrepancy through an increase in support for reparations for Black Americans.

The above analysis sets the stage for our central hypothesis. We tested this hypothesis in two preregistered studies:

> Hypothesis I: Exposure to information about redress for Japanese incarceration, versus a control condition, will elicit increased support for reparations for Black Americans.

Learning about Japanese redress is critical for creating the conditions for moral hypocrisy, which we expect to be necessary for changes in support for Black reparations. Thus, we also explored a second hypothesis, which we tested by examining responses to a quiz about Japanese redress administered at the end of each study:

> Hypothesis II: The extent of learning about Japanese redress in the intervention condition will account for the intervention's impact on support for Black reparations.

In addition to our attempts to measure how reminders of Japanese redress shape attitudes toward Black reparative economic justice, we also measured other constructs that might also explain why the intervention changed Asian American attitudes. In particular, we assessed whether reminders of Japanese redress, because they highlight shared experiences between Asian Americans and other marginalized populations, increase feelings of common fate between Asian and Black Americans (e.g., the success of Asian people depends on the success of Black people), reduce anti-black attitudes, increase perceptions of network diversity, reduce conservatism, and reduce internalization of model minority stereotypes (e.g., Asian Americans work harder than other groups). Across each of these constructs, we explored whether the intervention shifted endorsement of these attitudes, as well as, whether these attitudes moderated the impact of the intervention.

## Method

In our studies, we asked a sample of Asian Americans residing in the US to take a 10-min survey where we assigned them to our intervention, reminding participants of Japanese incarceration and redress, or to an informational control condition. Following the intervention, we asked participants to answer questions about reparations for Black Americans and some questions about political attitudes relevant to the present study. In Study 1, the control condition was an informational video depicting animals struggling with challenges created by climate change. In Study 2, the control condition was an informational video of Japanese American incarceration, without mention of the redress that followed. This control condition was chosen for Study 2 to rule out an alternative interpretation of our results in Study 1—that reminders about the injustice faced by Japanese Americans through their relocation and incarceration, and not the information about redress, changed participant attitudes toward Black reparations.

**Participants: Study 1**. We collected a sample of 329 Asian Americans from a variety of ethnic origin subgroups and generations in the US for an online study through Centiment Survey Panels, which recruits participants though online social networks for brief academic and marketing surveys. Respondents were paid $5 for a 10-min survey and all participants consented to participate in the study which was approved by the institutional review board at Yale University. Asian subgroups included Chinese (91), Indian (55), Filipino (54), Japanese (30), Korean (24), Vietnamese (23), Taiwanese (10), and the remaining participants indicated membership in one of the remaining 19 Asian subgroups (i.e., Bhutanese, Bangladeshi, Burmese, Cambodian, Fijian, Hmong, Hong Konger, Indonesian, Malaysian, Mongolian, Native Hawaiian, Nepali, Pakistani, Samoan, Singaporean, Sri Lankan, Taiwanese, Thai, Tongan). In terms of U.S. immigration generational status, most of our sample was first (166) or second (113) generation followed by third (28), fourth (11), or fifth (3) generation. Participants mostly self-identified as women (163), with 153 men, and 1 person identifying with another gender identity. A majority (181) of participants were currently employed, college graduates (248), and the median household income was $60,001–$80,000. Participants were 43.80 years of age on average ($SD = 18.21$). Table 1 displays a summary of this demographic information.

We decided to collect at least 300 participants for our intervention based primarily on budgetary considerations for the project. A sample of 329 Asian Americans allows us to detect an effect size Cohen's $d = 0.310$ with 80% statistical power. We report how we determined our sample size, all data exclusions (we did not make any), all manipulations, and all measures in the study.

**Participants: Study 2**. We collected a sample of 500 Asian Americans from a variety of ethnic origin subgroups and generations in the US for an online study through Prolific Academic, a sign-up based online crowdsourced survey response platform with a reputation for high data quality[25,26]. We used this alternative online sample to examine if our findings may generalize to this new group of Asian American participants. Respondents were paid $3 for a 10-min survey and all participants consented to participate in the study which was approved by the institutional review board at Yale University. Asian subgroups included Chinese (146), Indian (61), Filipino (55), Korean (56), Vietnamese (76), Japanese (30), Taiwanese (26), and the remaining participants indicated membership in one of the remaining 19 subgroups. In terms of American generational status, most of our sample was first (138) or second (326) generation followed by third (12), fourth (18), or fifth (6) generation. Participants were mostly self-identified as men (307), with 186 women, and 4 people identifying with another gender identity. A majority (309) of participants had college graduation as their highest level of education, and the median household income was $80,001–$100,000. Participants were 31.98 years of age on average

($SD = 9.80$). Table 1 displays a summary of this demographic information.

We decided to collect at least 500 participants for our intervention based primarily on budgetary considerations for the project. A sample of 500 Asian Americans allows us to detect an effect size Cohen's $d = 0.251$ with 80% statistical power. Importantly, this sample size also gives us more than 80% power to detect the intervention effect on reparations support observed in Study 1. We report how we determined our sample size (above), all data exclusions (we did not make any), all manipulations, and all measures in the study.

**Procedure: Study 1**. In the 10-min survey, participants first consented to participate in the research and confirmed that they self-identified as Asian American. We then randomly assigned each participant to an informational video about Japanese redress or a control video about nature which we describe in detail below and which lasted for a little more than two minutes each. Following the video, participants filled out their attitudes about reparations, and then they turned to a series of questions about their demographic and political information. After completing the study measures participants in both conditions were told about the design and intentions of the study and offered a chance to view the informational video of Japanese redress. At this time, participants had a chance to provide feedback for the study before they received payment for participation. All analyses that are preregistered are noted specifically in the results as such, whereas all other analyses are exploratory[27].

To inform participants about Japanese redress following incarceration we exposed participants to a 143-s informational video or a 136-s nature video control. In the informational video, the first author narrates a slide show describing the conditions around Japanese incarceration during World War 2, along with the struggle for redress that occurred in the decades to follow that culminated in the signing of the Civil Liberties Act of 1988. The video has slide images depicting life in the incarceration camps, and then shows Japanese Americans advocating with politicians for reparative federal action. As in the recommendations from social science research on reducing defensiveness to mentions of racial inequality, the intervention discusses the magnitude and impact of incarceration directly, and then frames that incarceration as a violation of shared American values around pursuit of the American Dream. Throughout the video, participants receive accompanying text transcripts of the audio as well as the opportunity to type out any feedback or reactions they have to the content of the video. This was our way to reduce defensiveness in responses to the informational intervention[8,9]. Most respondents provided some written feedback which we analyzed for Study 2 but not in Study 1, due to the topic differences between the intervention and control conditions.

The control video was chosen based on a few key considerations that we sought to keep consistent between the control and intervention conditions. We sought a video with a similar length, affect, and authority of the speaker to provide experimental conditions that help us rule out these as plausible alternative mechanisms to our findings. This is why, for the control video, we chose a nature video depicting mammals dealing with difficult climate conditions, which was similar in length, affect, and narrated by an apparent expert with accompanying transcripts. See Supplementary Methods for analyses of features of the intervention and control conditions.

**Procedure: Study 2**. In the 10-min survey, participants first consented to participate in the research and confirmed that they self-identified as Asian American. We then randomly assigned

each participant to an informational video about Japanese incarceration and redress or a control video about incarceration only. Following the video, participants filled out their attitudes about reparations, and then they turned to a series of questions about their demographic and political information as in Study 1. As in Study 1, all analyses that are preregistered are noted specifically in the results as such[28].

The intervention was identical to Study 1, informing participants about incarceration and redress payments for Japanese Americans. The control condition was identical to the first half of the intervention condition, except that the control condition was titled, "the story of Japanese incarceration" rather than "the story of Japanese reparations." As in the prior study, we reduced defensiveness by giving participants the opportunity to type out any feedback or reactions they have to the content of the video. Most respondents provided some written feedback which we explore using qualitative and quantitative text analyses in Supplementary Note 1.

**Measures: Study 1**. Participants answered all questions about reparations directly following the manipulation. The questions were adapted from national panel surveys of attitudes toward Black reparations[19]. The items were, "Do you think the federal government should or should not make cash payments to communities in the USA for past harms?", "Do you think the federal government should or should not make cash payments to the descendants of slaves?", and "Do you think the federal government should or should not establish a commission to study the effects of slavery and recommend potential remedies?". A fourth item assessed perceptions of the feasibility of reparations: "I believe that reparations for Black Americans will or will not happen in my lifetime." Each item was assessed on a four-point scale with scale anchors of 4 = *definitely should/will*, 3 = *probably should/will*, 2 = *probably should not/will not*, 1 = *definitely should not/will not*. To assess support for each policy we first analyze all these items separately. Interestingly, data analysis revealed that the feasibility item about whether or not reparations will happen in my lifetime was not correlated highly with the other items ($\alpha = 0.56$) and so in our exploratory analyses we use an overall composite across the first three items reflecting policy support ($\alpha = 0.82$). In all cases, responses were coded so that higher scores indicate greater support for reparations ($M = 2.81$, $SD = 0.79$).

We additionally asked participants to complete a series of political attitude scales that we expected to be related to the topic of this study. We assessed the diversity of participant social networks using four items of self-assessed diversity (e.g., "Are all the people you typically interact with in your current neighborhood") using a four point scale (1 = *all the same race as you*, 2 = *mostly the same*, 3 = mostly different, 4 = *all of a different race*; $M = 2.98$, $SD = 0.66$, $\alpha = 0.77$) from prior research[29]. We assessed social and economic conservatism on two items as in prior research, "I am a conservative when it comes to social/economic issues" (1 = *strongly disagree*, 7 = *strongly agree*; $M = 3.86$, $SD = 1.56$, $\alpha = 0.85$). Interestingly, the Study 1 sample tended to be slightly more conservative than prior research on the politics of Asian Americans would suggest[30].

We also assessed common fate with Black Americans using a single item measure, "How much does Asian Americans doing well depend on Black Americans also doing well" (1 = *Not at all*, 7 = *A lot*; $M = 3.64$, $SD = 1.98$) from prior research[6]. We assessed a feeling thermometer measure of positive or negative attitudes toward Black ($M = 71.20$, $SD = 24.21$) and Asian Americans ($M = 82.76$, $SD = 17.09$) using a 0–100 scale where higher scores indicate more positivity. We assessed model minority stereotype

beliefs related to work ethic (e.g., "Asians have stronger work ethic."; $M = 5.25$, $SD = 1.60$, $\alpha = 0.94$) and exposure to discrimination (e.g., "Asians are less likely to experience racism in the United States."; $M = 3.23$, $SD = 1.59$, $\alpha = 0.88$), ($1 = strongly$ $disagree$, $7 = strongly$ $agree$) using methods from prior research[15]. See Supplementary Note 2 for analyses examining these beliefs.

Interspersed among the above items was a single quiz question to assess whether participants paid attention to the intervention video on Japanese redress. The question asked if Japanese Americans had received cash payments from the government, and if so, for how much? The response options were "no," "yes, $20,000" (the correct response), and "yes, $100,000".

**Measures: Study 2**. Participants answered all questions about reparations directly following the manipulation. The questions were identical to those used in Study 1. We used the same three item composite to index support for Black reparations ($\alpha = 0.86$). In all cases, responses were coded so that higher scores indicate greater support for reparations ($M = 2.99$, $SD = 0.79$).

We again measured participants' political attitudes using the same items as in Study 1 which included: diversity of participant social network ($M = 2.88$, $SD = 0.53$, $\alpha = 0.69$), social and economic conservatism ($M = 2.97$, $SD = 1.57$, $\alpha = 0.80$), common fate with Black Americans ($M = 3.28$, $SD = 1.58$, $\alpha = 0.94$), positive or negative attitudes toward Black ($M = 64.19$, $SD = 24.12$) and Asian Americans ($M = 80.44$, $SD = 17.34$), model minority stereotype beliefs related to work ethic ($M = 4.90$, $SD = 1.50$, $\alpha = 0.93$) and model minority stereotype beliefs related to exposure to discrimination ($M = 3.44$, $SD = 1.50$, $\alpha = 0.91$).

We also assessed anti-Black attitudes using a 10-item scale ($M = 2.68$, $SD = 0.92$, $\alpha = 0.91$)[31]. Finally, two items about felt similarity to Black Americans were also added to the survey, based on prior research finding associations between similarity and common fate judgments (e.g., "I think I am similar to many Black people")[6]. These items created a similarity composite ($M = 3.46$, $SD = 1.48$, $\alpha = 0.92$). Interspersed among the above items was the same quiz question to assess whether participants learned more about Japanese redress in the intervention versus the control condition. Data distributions were visually inspected for normality, and assumed to be normal based on this inspection, prior to formal hypothesis testing, but this was not formally tested[32].

## Results

**Learning about Japanese redress**. If our intervention is likely to shape attitudes toward reparations, a preliminary step is that it must first inform Asian American participants about the existence of Japanese redress payments. In Study 1, we expected and found that participants answered the quiz question correctly more in the intervention condition (74.1%) than the control condition (45.9%), $X^2(1) = 26.95$, $p < .001$, indicating that our manipulation was successful in informing participants about Japanese redress. Moreover, when we recoded quiz responses to give correct scores to responses that also included answers where the amount paid was incorrect, but answered that the redress payments had occurred, the proportion of correct responses in the intervention condition rose further to 80.4%.

The same analysis in Study 2 yielded similar results: We expected and found that participants answered the quiz question correctly more in the intervention condition (91.9%) than in the control condition (44.8%), $X^2(1) = 127.82$, $p < 0.001$, indicating that our manipulation was successful in informing participants about Japanese redress. Recoding quiz responses to include partial credit led the proportion of correct responses in the intervention

condition to rise further to 93.5%. See Supplementary Note 3 for preliminary tests showing the success of our random assignment manipulation.

**Intervention and support for Black reparations**. Our central prediction was that the informational intervention about Japanese redress would increase support for reparations for Black Americans relative to the control condition. We conducted this pre-registered analysis by comparing means between the intervention and control conditions for each of the four distinct items about support for reparations and the composite measure.

In Study 1, some results were consistent with our preregistered predictions: For two of the four items, support for reparations was significantly higher in the intervention condition than the control condition. This included the first item about general support for cash payments from the government for past injustice ($M_{intervention} = 3.06$, $SD_{intervention} = 0.85$; $M_{control} = 2.76$, $SD_{control} = 0.88$, $t(327) = 3.116$, $p = 0.002$, $d = 0.344$, 95% Confidence Intervals of mean difference $= 0.110$ to $0.486$, and support for Black reparations for descendants of enslaved people ($M_{intervention} = 2.68$, $SD_{intervention} = 0.96$; $M_{control} = 2.45$, $SD_{control} = 0.95$), $t(327) = 2.146$, $p = 0.033$, $d = 0.237$, 95% Confidence Intervals $= 0.019$ to $0.434$. We found no statistically significant evidence of a difference for commissioning a federal study of the cost and impacts of slavery ($M_{intervention} = 3.03$, $SD_{intervention} = 0.92$; $M_{control} = 2.88$, $SD_{control} = 0.95$), $t(327) = 1.499$, $p = 0.135$, $d = 0.165$, 95% Confidence Intervals $= -0.048$ to $0.358$. The overall composite across these three items on reparations policy also showed greater support for reparations in the intervention than the control condition ($M_{intervention} = 2.92$, $SD_{intervention} = 0.76$; $M_{control} = 2.70$, $SD_{control} = 0.81$), $t(327) = 2.616$, $p = 0.009$, $d = 0.289$, 95% Confidence Intervals $= 0.056$ to $0.397$ (Fig. 1). The final feasibility item about whether reparations would happen in my lifetime showed the opposite pattern where participants in the control condition ($M = 2.41$, $SD = 0.85$) thought reparations were more likely to happen than in the intervention condition ($M = 2.18$, $SD = 0.72$), $t(327) = -2.716$, $p = 0.007$, $d = 0.300$, 95% Confidence Intervals $= 0.065$ to $0.406$.

Study 2 results were also consistent with our preregistered predictions: Support for reparations was significantly higher in the intervention condition than the control condition for our three items about support for reparations. This included the first item about general support for cash payments from the government for past injustice ($M_{intervention} = 3.20$, $SD_{intervention} = 0.77$; $M_{control} = 2.87$, $SD_{control} = 0.88$), $t(498) = 4.383$, $p < 0.001$, $d = 0.392$, 95% Confidence Intervals of mean difference $= 0.179$ to $0.470$, support for Black reparations for descendants of enslaved people ($M_{intervention} = 2.84$, $SD_{intervention} = 0.95$; $M_{control} = 2.63$, $SD_{control} = 0.95$), $t(498) = 2.481$, $p = 0.013$, $d = 0.222$, 95% Confidence Intervals $= 0.044$ to $0.80$, and support for commissioning a federal study of the cost and impacts of slavery ($M_{intervention} = 3.35$, $SD_{intervention} = 0.81$; $M_{control} = 3.08$, $SD_{control} = 0.94$), $t(498) = 3.566$, $p < 0.001$, $d = 0.319$, 95% Confidence Intervals $= 0.125$ to $0.433$. The overall composite across these three items also showed greater support for reparations in the intervention than the control condition ($M_{intervention} = 3.13$, $SD_{intervention} = 0.74$; $M_{control} = 2.86$, $SD_{control} = 0.82$), $t(498) = 3.898$, $p < 0.001$, $d = 0.349$, 95% Confidence Intervals $= 0.135$ to $0.409$ (Fig. 1). We found no statistically significant evidence of an intervention difference on whether Black American reparations would happen in my lifetime between the control condition ($M = 2.19$, $SD = 0.66$) or the intervention condition ($M = 2.27$, $SD = 0.71$), $t(498) = 1.169$, $p = 0.243$, $d = 0.105$, 95% Confidence Intervals $= -0.192$ to $0.049$.

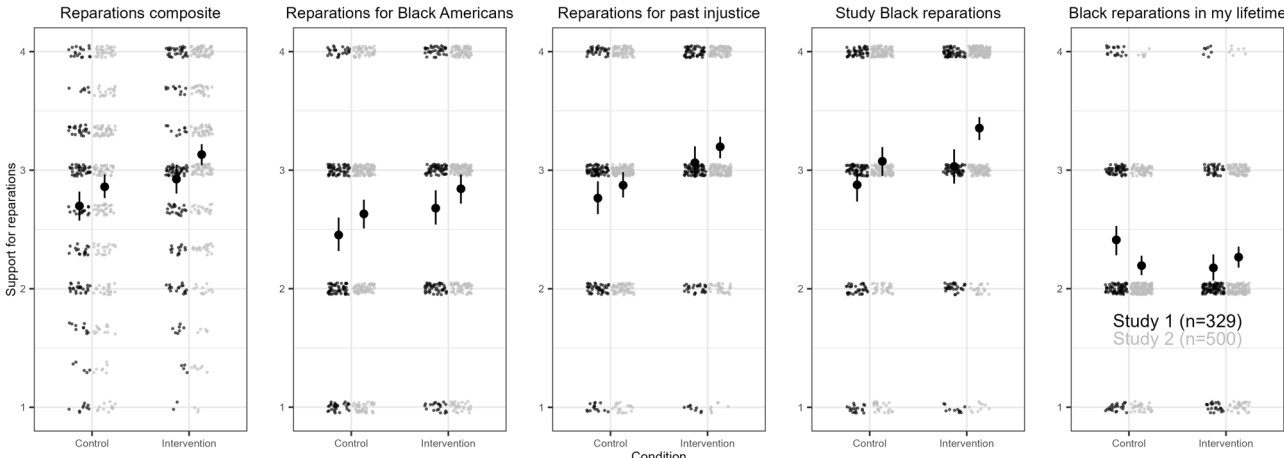

**Fig. 1 Support for Black reparations among Asian American participants randomly assigned to the control or intervention conditions.** Responses in Study 1 are represented in black and Study 2 data are in gray. Higher scores indicate more support for reparations policies or that participants thought reparations would be more likely (versus less likely) to happen in their lifetime. The composite measure is an average of the first three policy items. Smaller individual dots indicate jittered participant responses to each question and larger black dots with brackets indicate means and 95% confidence intervals surrounding the mean.

**Exploratory analyses on learning and support for Black reparations.** To better understand whether learning about Japanese redress, versus some other shift in attitudes, explains greater support for reparations for Black Americans, we used Process Model 4[33], to conduct an exploratory mediation analysis with reparations support as the dependent variable, intervention as the interdependent variable, and responses to the quiz question about Japanese redress as the mediator. Because Process does not accept dichotomous mediators, quiz responses were rescored so that answering "yes, $100,000" was rescored as "1," a partially correct response, while "yes, $20,000" was rescored as "2," or a fully correct response, and "no" was rescored as "0," or an incorrect response. In this analysis, we controlled for demographic variables related to education and gender. The latter variables were added to the model because education is confounded with knowledge, and gender was associated with reparations support in Study 2. Analyses without these controls were consistent with those reported here.

In Study 1, the model found a significant effect of the intervention on the quiz mediator $B = 0.605$ (0.101), $t(311) = 5.99$, $p < 0.001$, a significant effect of the quiz mediator on reparations support $B = 0.109$ (0.049), $t(310) = 2.22$, $p = 0.0274$, and a significant effect of the intervention on reparations support in the same model $B = 0.195$ (.093), $t(310) = 2.11$, $p = 0.0357$. Bootstrapping analysis with 5000 resamples revealed a significant indirect effect of the intervention on reparations support through the quiz mediator $B = 0.066 (0.0318)$, 95% Confidence Intervals = 0.008 to 0.135. In the model, there was no statistically significant evidence of a relationship between education and reparations support, $B = -0.0415$ (0.061), $t(310) = -0.680$, $p = 0.4973$, nor between gender and reparations support, $B = -0.166$ (0.061), $t(310) = -1.887$, $p = 0.060$.

In Study 2, the model found a significant effect of the intervention on the quiz mediator $B = 0.931$ (0.070), $t(496) = 13.305$, $p < 0.001$, a significant effect of the quiz mediator on reparations support $B = 0.133$ (0.044), $t(495) = 3.037$, $p = 0.0025$, and a significant effect of the intervention in the same model $B = 0.161$ (.080), $t(495) = 2.0247$, $p = 0.0434$. Bootstrapping analysis with 5000 resamples revealed a significant indirect effect of the intervention on reparations support through the quiz mediator $B = 0.1611$ (0.0433), 95% Confidence Intervals = 0.041 to 0.211. In the model, there was no statistically

significant evidence of a relationship between education and reparations support, $B = 0.024$ (0.056), $t(495) = 0.4200$, $p = 0.6747$. There was a significant relationship between gender and reparations support, $B = -0.267$ (0.071), $t(495) = -3.7769$, $p < 0.001$. Overall, we provide some statistical evidence consistent with our assertion that higher support for Black reparations in the intervention condition was statistically accounted for by increased knowledge of Japanese redress payments in the intervention, relative to the control condition.

Study 1 correlations between our political attitude measures and reparations composite are shown in Table 2. As you can see from the table, higher support for our reparations composite was associated with lower network diversity, which we interpret as greater contact with other Asian Americans, lower conservatism, lower endorsement of the work ethic component of internalized model minority stereotypes, higher judgments of common fate with Black Americans, more positive feelings toward Black and Asian Americans, and lower income. Correlations between our political attitude measures collected in Study 2 and the reparations composite are shown in Table 3. As you can see from the table, higher support for our reparations composite was associated with lower conservatism, lower endorsement of the work ethic component of internalized model minority stereotypes, higher judgments of common fate with Black Americans, greater similarity to Black Americans, less anti-Black attitudes, more positive feelings toward Black and Asian Americans, and women and non-binary individuals reported more support for reparations than men. See Supplementary Note 2 for tests of the intervention on participant political attitudes which were not statistically significant in all cases except in Study 2, where Asian American participants reported that they faced fewer barriers in the intervention condition $(M = 3.59)$ than in the control condition $(M = 3.29)$, $t$ (498) = 2.239, $p = 0.026$, 95% Confidence Intervals = 0.037 to 0.561. It is possible, that information about Japanese Americans winning redress, where Black Americans had not, increased a belief in lower barriers experienced by Asian Americans. We interpret this finding with caution, though, given that in the pooled sample no statistically significant evidence for intervention differences in internalized model minority stereotypes emerged. Overall, these analyses indicate that we found no evidence that our intervention shaped support for reparations through broader political attitude change.

**Table 2 Correlations between political attitudes, education, income, gender, and support for reparations payments in Study 1.**

| | | 1 | 2 | 3 | 4 | 5 | 6 | 7 | 8 | 9 | 10 |
|---|---|---|---|---|---|---|---|---|---|---|---|
| 1 | Reparations | --- | | | | | | | | | |
| 2 | Network diversity | −0.14* | --- | | | | | | | | |
| 3 | Conservatism | −0.32** | −0.03 | --- | | | | | | | |
| 4 | Model minority work ethic | −0.23** | −0.09 | 0.32** | --- | | | | | | |
| 5 | Model minority discrimination | −0.02 | −0.10 | 0.14* | 0.17** | --- | | | | | |
| 6 | Common fate | 0.36** | −0.10 | −0.09 | −0.09 | 0.20** | --- | | | | |
| 7 | Feel for Black Americans | 0.30** | 0.09 | −0.15** | −0.20** | 0.00 | 0.30** | --- | | | |
| 8 | Feel for Asian Americans | 0.20** | −0.01 | −0.19** | 0.11 | −0.12* | 0.08 | 0.56** | --- | | |
| 9 | Household income | −0.16** | −0.02 | −0.02 | 0.08 | −0.07 | −0.16** | −0.02 | 0.04 | --- | |
| 10 | Education | −0.04 | −0.09 | 0.07 | 0.10 | 0.06 | −0.09 | −0.08 | −0.02 | 0.24** | --- |
| 11 | Gender | −0.09 | 0.01 | 0.03 | 0.08 | 0.09 | −0.06 | −0.12* | −0.05 | 0.14* | 0.01 |

Asterisks indicate that $p < 0.05$ (*) and $p < 0.01$ (**).

See Supplementary Note 2 for analyses of exploratory interaction effects between the intervention and our political attitudes and demographic measures aggregated across our two studies. Overall, we did not find credible evidence for moderation of the intervention as a function of conservatism, common fate beliefs, internalized model minority stereotypes, network diversity, or whether participants were Japanese Americans or not. We found evidence for an interaction between generational status and the intervention while controlling for the confound of age. In this interaction, earlier generations of Asian Americans tended to be more likely to support reparations than later generations in the intervention condition, whereas no statistically significant evidence of an effect of generation occurred in the control condition. See Supplementary Note 2 for these analyses, which could be related to knowledge of Japanese redress being lower among Asian Americans who more recently immigrated.

One of the primary goals of this research is to better understand the extent that informational interventions such as this one about Japanese incarceration and redress might increase support for reparative economic action for Black Americans. One way to assess the overall efficacy of the intervention is to assess the overall size of the effect of the intervention across our two studies[34]. To determine the effect size of the intervention, we conducted a mini-meta analysis across our two studies by pooling the data from our two experiments into a single analysis which then treats study as a fixed factor. See Supplementary Note 4 for analyses supporting this pooling decision.

The pooled analysis shows an overall increase in support for reparations payments as a result of the informational intervention versus the control conditions $F(1,825) = 20.145$, $p < 0.001$, $\eta_p^2 = 0.024$, $d = 0.313$, 95% Confidence Intervals = 0.057 to 0.396. There was also a significant study effect where higher support for reparations came in general from Study 2 regardless of condition, presumably because participants were more liberal in that study $F(1,825) = 11.04$, $p < 0.001$, $\eta_p^2 = 0.013$, 95% Confidence Intervals = 0.051 to 0.363. No interaction between study and condition emerged $F(1,825) = 0.167$, $p = 0.682$, $\eta_p^2 < 0.001$, 95% Confidence Intervals = −0.173 to 0.263.

**National polling comparisons.** Another way to interpret the effectiveness of the intervention is to compare support for Black reparations across our studies to support for reparations observed in nationally representative opinion polls on the topic. To conduct this comparison we recoded answers to the single item on preferences for Black reparations such that responses either reflected being in favor of, or against, reparations, where responses were recoded as either in favor or opposed to Black reparations. We then compared the proportion of those supporting Black reparations payments in our control and intervention conditions pooled across our two studies to available national polls which date back to 2014. The results of one sample t-tests comparing our intervention condition to these polls shows that the intervention significantly increased support for Black reparations payments relative to all comparison polling data, including an opinion survey of Asian Americans conducted in 2021, $t(421) = 4.784$, $p < 0.001$, $d = 0.237$, 95% Confidence Intervals = 0.067 to 0.161 (Fig. 2). In contrast, the control condition showed no statistically significant differences in comparison to support for reparations among a subsample of Asian Americans conducted at UMass in 2021 $t(421) = 0.717$, $p = 0.474$, $d = 0.035$, 95% Confidence Intervals = −0.030 to 0.065.

Figure 2 also allows us to better understand and interpret the magnitude of the intervention's impact on support for reparations for Black Americans. Though an effect size of $d = 0.313$ ($\eta_p^2 = 0.024$) is modest, effects need not be large to be important[35,36]. If we compare the difference in reparations support between the control and intervention across our studies we see that there is nearly a 10% difference in support for reparations payments, which is roughly equivalent to 46.7% of the total progressive attitude change in reparations support from 2014 to 2021.

## Discussion

The relocation and incarceration of Japanese Americans was a grave injustice visited on more than 120,000 people, and though the US government did eventually offer apology and monetary redress for this injustice, these actions are not well-known. In this study, we examined the capacity for an informational intervention about the history of Japanese American incarceration and the struggle for redress to increase Asian American support for reparative economic action for another marginalized population —Black Americans seeking justice for chattel slavery and Jim Crow. Results indicate that the informational intervention was more effective than a control condition for increasing support for this kind of redress (Hypothesis I). In fact, the levels of support of reparations for Black Americans observed following our intervention were higher than the support measured in public opinion surveys of reparations collected between 2014 and when this study was fielded.

Importantly, we observed the success of our intervention informing about Japanese American redress relative to an informational control about animals struggling with climate change and one that described Japanese incarceration (but not redress) in identical terms to the intervention. The latter finding with the

**Table 3 Correlations between political attitudes, education, income, gender, and support for reparations payments in Study 2.**

| | 1 | 2 | 3 | 4 | 5 | 6 | 7 | 8 | 9 | 10 | 11 | 12 |
|---|---|---|---|---|---|---|---|---|---|---|---|---|
| 1 Reparations | --- | | | | | | | | | | | |
| 2 Network diversity | −0.02 | --- | | | | | | | | | | |
| 3 Conservatism | −.511** | 0.00 | --- | | | | | | | | | |
| 4 Model minority worth ethic | −.362** | −.099* | .374** | --- | | | | | | | | |
| 5 Model minority discrimination | 0.05 | −.088* | −0.01 | −0.04 | --- | | | | | | | |
| 6 Common fate | .481** | 0.03 | −.207** | −.334** | 0.01 | --- | | | | | | |
| 7 Similar to Black Americans | .434** | .126** | −.238** | −.342** | −0.03 | .766** | --- | | | | | |
| 8 Anti-Black attitudes | −.569** | −0.05 | .539** | .630** | −0.01 | −.465** | −.464** | --- | | | | |
| 9 Feel for Black Americans | .407** | .139** | −.248** | −.395** | 0.07 | .485** | .520** | −.548** | --- | | | |
| 10 Feel for Asian Americans | .241** | 0.05 | −.103* | −0.02 | 0.01 | .186** | .177** | −.198** | .561** | --- | | |
| 11 Household income | 0.00 | −0.02 | 0.03 | −0.03 | 0.04 | −0.03 | −0.04 | −0.01 | −0.04 | 0.00 | --- | |
| 12 Education | 0.02 | −0.05 | 0.00 | −0.02 | 0.02 | 0.06 | 0.05 | −.091* | 0.00 | 0.04 | .256** | --- |
| 13 Gender | −.168** | −0.04 | .190** | .110* | .101* | −.124** | −.090* | .251** | −.107* | −0.05 | −0.04 | −0.08 |

Asterisks indicate that $p < 0.05$ (*) and $p < 0.01$ (**).

incarceration only control condition suggests that it was the reminders of Japanese redress, in particular, that increased support for reparations for Black Americans. Moreover, that the effect of the intervention was consistent across the two studies despite significant differences in conservatism and mean level support for reparations across the two samples is informative about the generalizability of the effect of the intervention to more and less conservative Asian American samples.

A few exploratory analyses provide some clues about the psychological processes that made our intervention successful. Across our studies, there was no statistically significant evidence that the informational intervention changed Asian American participants' broader political attitudes around general conservatism, feelings toward Black and Asian Americans, common fate with Black Americans, or endorsement of model minority stereotypes. Instead, exploratory mediation analyses in each of the two studies indicate that learning about redress of a subgroup of Asian Americans closely related to one's own group (i.e., Japanese Americans) increased support for redress for Black Americans (Hypothesis II). The evidence was consistent with participants avoiding hypocrisy in their responses[5]: That is, the significant increased knowledge of the success of Japanese redress in the intervention condition made general support for redress for another marginalized group more appealing than in the control condition where participants were not exposed to that knowledge.

The current study advances social scientific understanding in a few key respects. As in prior work, where reminders of past discriminatory experiences increase solidarity between marginalized groups[6,7], the current work helps us understand if knowledge and meaning making around the history of redress of one marginalized group is a means to increase support for reparative policies for other marginalized populations. Moreover, given the history of Asian Americans, and Japanese Americans in particular, as being initially resistant to discussing redress for incarceration[1], the negative impacts of incarceration on contemporary political engagement[37], and the historical and contemporary use of Asian populations as a political wedge model minority group against other marginalized populations[3], this research helps us better understand and reverse divisions between marginalized groups.

The study also helps clarify how informational interventions might shape policy attitudes. Interventions intended to teach an accurate history of American racism have mixed evidence with respect to their effectiveness in policy discussions[38]. This research aligns with past studies suggesting that an intervention that provides new information, and also takes steps to message in ways that reduce defensiveness can be effective in changing attitudes[8,9].

Interestingly, there was no statistically significant evidence that reminders of the Japanese American struggle for redress reliably changed people's conceptions of whether Black reparations were less, versus more likely in one's lifetime, relative to the control condition. Prior research indicates that Americans tend to be more optimistic about positive change toward racial equality than data trends would suggest[20], but perhaps that tendency is countered by being confronted with the political realities of the 45-year struggle for Japanese redress. More research is necessary to better understand theories of progress and change among Asian Americans and in the context of reparations specifically.

**Limitations.** The collection of groups that make up the umbrella of Asian Americans is a political alliance based on common goals and shared experiences in the US[30]. That an intervention focused on Japanese relocation, incarceration, and redress increased support for Black reparations among a diverse sample of Asian

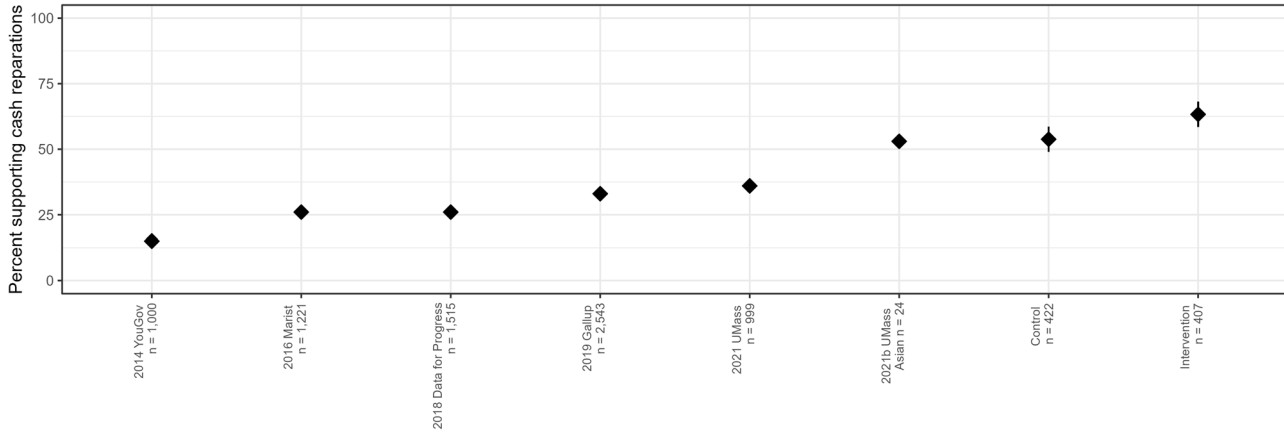

**Fig. 2 Percentage of polled participants who reported that they support cash payments for reparations paid to descendants of enslaved Black Americans across time and by polling source.** Error bars for the control and intervention conditions of our pooled data across the two studies represent 95% confidence intervals surrounding the mean.

Americans speaks broadly to the capacity of Asian American subgroups to experience solidarity across their various identities. That the intervention shaped support for reparations across immigrant generational status and whether participants identified as Japanese American versus from another Asian origin group is indicative of the potential for political solidarity among Asian communities. However, the current work here could be built upon with more extensive targeted sampling to better understand if certain Asian origin subgroups have more or less capacity for political solidarity.

A related question to the above is, would our intervention work for white Americans who might be motivated to support reparations for Black Americans when presented with information about Japanese redress? One line of reasoning suggests this intervention might be effective: If support for reparations is based on a need to avoid hypocrisy solely, then we might expect the intervention to work also for white Americans out of a desire to avoid redress for one group and not another[5]. However, if support for Black reparations is also based in stigma-based solidarity, then white Americans may not be persuaded by this particular message[7]. Future research is necessary to test these competing predictions.

This study was not without limitations that call for caution when interpreting these results beyond the online panels from which the studies were fielded. The Asian American category is a coalition of people from Asian origin countries with many unique immigration histories in the US and languages. As such, our online panel samples have failed to fully capture the complexity of this broad collection of people. A national probability sample is warranted to better understand how the intervention we used here shapes attitudes toward reparations for Black Americans. As we mentioned above, targeted sampling is also warranted, to better understand how specific groups, such as Japanese Americans in particular, might respond more or less strongly to the intervention. Likewise, that the studies were conducted in English limits our conclusions to Asian Americans who would be likely to seek out opportunities to complete surveys in English.

## Conclusions

The capacity to teach and to learn about the multi-racial history of the US is low according to recent analyses of the small number of ethnic studies programs in US higher education institutions[39]. Nevertheless, in this study, we show the importance of these

history lessons. Specifically, we show that Asian Americans who are more historically informed—about precedent for redressing past injustice through study, apology, and monetary compensation—might also be more likely to act in solidarity with other marginalized communities in the service of reparative economic policies. Understanding the ways to message about this history that galvanize solidarity, the pursuit of justice, and importantly, minimize backlash, remains an important area of future inquiry across the social sciences.

**Reporting summary**. Further information on research design is available in the Nature Portfolio Reporting Summary linked to this article.

## Data availability

All data is available for download and reanalysis on https://doi.org/10.17605/OSF.IO/MUPKC[40].

## Code availability

All code for reproducing the analysis and study materials for reproducing the experiments are located at https://doi.org/10.17605/OSF.IO/MUPKC[40]. All analyses were conducted on SPSS version 28.

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

## Acknowledgements

The authors wish to thank Anne Yamamoto Mow for making this scholarship possible. Time spent preparing this manuscript was supported by research grant #G-2111-34528 from the Russell Sage Foundation. The funder had no role in study design, data collection and analysis, decision to publish or preparation of the manuscript.

## Author contributions

M.W.K. designed the studies and collected the data. M.W.K. and A.C.V. wrote the paper, analyzed the data, and prepared critical revisions to the analysis and writing.

## Competing interests

The authors declare no competing interests.
