## [Peer review file · Communications Psychology]

3rd Aug 23

Dear Professor Kraus,

Thank you for your patience during the peer-review process. Your manuscript titled "Reminders of Japanese redress and Asian American support for Black reparations" has now been seen by 3 reviewers, whose comments are appended below. You will see that they find your work of some potential interest. However, they have raised quite substantial concerns that must be addressed. In light of these comments, we cannot accept the manuscript for publication, but would be interested in considering a revised version that fully addresses these serious concerns.

We hope you will find the Reviewers' comments useful as you decide how to proceed. Should additional work allow you to address these criticisms, we would be happy to look at a substantially revised manuscript. If you choose to take up this option, please highlight all changes in the manuscript text file, and provide a detailed point-by-point reply to the reviewers.

Editorially, we consider it important that the revised manuscript transparently discusses the limitations of the current methodology and restrains the conclusions drawn to those supported by the data. As you will see, the referees voiced significant concerns regarding the lack of representativeness of the sample. As you will see below, the referees' initial assessment as to whether this limitation precludes publication differed. Considering all arguments and after seeking additional referee feedback, we decided that on balance we do not consider representative samples necessary to interpret experimental effects; however, the sampling limitations significantly decrease the potential to draw inferences onto the general population. We ask you to keep this in mind for the interpretation of follow-up analyses and make it clear in the Abstract and the "Limitations" section that forms part of the Discussion that the findings cannot be extrapolated to Asian Americans in general. We also encourage you to consider the broader context of Asian American-Black American intergroup relations as suggested by Reviewers 1 and 3.

For manuscripts that report null results, we require Bayes Factors or equivalence tests to interpret the null results appropriate language to describe the results. (There is no statistical test that can demonstrate absence of an effect. Statements such as 'There is no difference between x and y.' or 'X does not affect Y.' must be revised to read 'We found [no/little] credible evidence of a difference between x and y.' or 'We found [no/little] credible evidence that X affects Y.')

Please remove claims of significance for any results with p values higher than .05. These results must be reported as non-significant.

If the revision process takes significantly longer than five months, we will be happy to reconsider your paper at a later date, provided it still presents a significant contribution to the literature at that stage.

Please use the following link to submit your revised manuscript, point-by-point response to the Reviewers' comments with a list of your changes to the manuscript text (which should be in a separate document to any cover letter) and any completed checklist:

[link redacted]

Please do not hesitate to contact me if you have any questions or would like to discuss the required revisions further. Thank you for the opportunity to review your work.

Best regards,

Jennifer Bellingtier

Jennifer Bellingtier, PhD
Senior Editor
Communications Psychology

EDITORIAL POLICIES AND FORMATTING

Editorial Policy: [Policy requirements](https://www.nature.com/documents/nr-editorial-policy-checklist.pdf) (Download the link to your computer as a PDF.)

Furthermore, please align your manuscript with our format requirements, which are summarized on the following checklist:

[Communications Psychology formatting checklist](https://www.nature.com/documents/commspsychol-style-formatting-checklist-article-rr.pdf)

and also in our style and formatting guide [Communications Psychology formatting guide](https://www.nature.com/documents/commspsychol-style-formatting-guide-accept.pdf) .

* TRANSPARENT PEER REVIEW: Communications Psychology uses a transparent peer review system. This means that we publish the editorial decision letters including Reviewers' comments to the

authors and the author rebuttal letters online as a supplementary peer review file. However, on author request, confidential information and data can be removed from the published reviewer reports and rebuttal letters prior to publication. If your manuscript has been previously reviewed at another journal, those Reviewers' comments would not form part of the published peer review file.

* **CODE AVAILABILITY:** All Communications Psychology manuscripts must include a section titled "Code Availability" at the end of the methods section. In the event of publication, we require that the custom analysis code supporting your conclusions is made available in a publicly accessible repository; please choose a repository that provides a DOI for the code; the link to the repository and the DOI must be included in the Code Availability statement. Publication as Supplementary Information will not suffice. We ask you to prepare and upload code at this stage, to avoid delays later on in the process.

* **DATA AVAILABILITY:**

All Communications Psychology research manuscripts must include a section titled "Data Availability" at the end of the Methods section or main text (if no Methods). More information on this policy, is available at <http://www.nature.com/authors/policies/data/data-availability-statements-data-citations.pdf>.

At a minimum the Data availability statement must explain how the data can be obtained and whether there are any restrictions on data sharing. Communications Psychology strongly endorses open sharing of data. If you do make your data openly available, please include in the statement:

We recommend submitting the data to discipline-specific, community-recognized repositories, where possible and a list of recommended repositories is provided at <http://www.nature.com/sdata/policies/repositories>.

If a community resource is unavailable, data can be submitted to generalist repositories such as [figshare](https://figshare.com/) or [Dryad Digital Repository](http://datadryad.org/). Please provide a unique identifier for the data (for example a DOI or a permanent URL) in the data availability statement, if possible. If the repository does not provide identifiers, we encourage authors to supply the search terms that will return the data. For data that have been obtained from publicly available sources, please provide a URL and the specific data product name in the data availability statement. Data with a DOI should be further cited in the methods reference section.

REVIEWER EXPERTISE:

Reviewer #1 Intergroup Relations

Reviewer #2 Political Psychology

Reviewer #3 Minority Groups, Intergroup Relations

REVIEWERS' COMMENTS:

Reviewer #1 (Remarks to the Author):

I appreciate the opportunity to review this article. In general, this article finds that reminding Asian Americans of the Japanese redress of Incarceration would increase Asian Americans' support for the redress of African American for slavery, and such effects are not through the change of political attitudes, perception of linked-fate, or anti-blackness.

I think this paper can make a great contribution to the study of Asian Americans as well as the study of public policy opinions. I think this paper has great potential and is well-written. I hope to make the following comments and suggestions.

First, I would encourage the author to have more discussions on Asian American and their attitudes on inter-racial relations with African American and/or other minorities. As the author points out in p.29, there might be differences between Asian American and white Americans regarding the intervention effect. It makes sense given that, being a minority themselves, Asian Americans may have different opinions on how other minority groups (in this paper, African American) should be treated. Some recent studies and the Harvard Lawsuit seem to indicate that Asian Americans are not necessarily supportive of policies that benefit other minority groups, especially when they think those policies are benefiting other groups at the expense of Asian American interests. I believe engaging with the attitudes of Asian American toward other minority groups would be important for this paper.

Second, I am a little unsure about the mechanism this paper tries to demonstrate. It seems acceptable to say that the treatment effects are not caused by any change in political attitudes or perceptions of African American, but what exactly is the mechanism? I feel it is a little weak to say the mechanism is "solution aversion" or "avoiding hypocrisy." I would encourage more discussions and (if possible) more analysis on this.

Third, I am not sure I fully understand Table 1 and the relevant discussions (also the same discussions in Study 2). Things like their level of conservatism and their network diversity should not be changed by the treatment material. Thus, if the experimental randomization is successful, we should see a balance across these items that are not changed by treatment. This means the level of conservatism and network diversity should be similar in the control and treatment groups. But if so, how can they be correlated with the support of reparation, which is clearly correlated with experimental groups?

Fourth, the design of Study 2 can only show that Japanese redress itself can have some effects, but it cannot rule out that knowing the unfair treatment of Japanese also leads to increased support of Black reparations because no pure control groups are presented in experiment 2. Why is it designed in this way? Also, I think the author should be careful in claiming that only the Japanese reparation

message works because we actually do not know whether the Japanese Incarceration messages also work or not.

Fifth, I don't feel the text analysis section is sufficiently convincing. It is natural to see more people write about getting justice in treatment groups in study 2 because that's what the treatment group's message is about. Similarly, for the quantitative analysis of texts, what can we conclude from the fact that "use of acquire related words were associated with support for reparations"? As said, the usage of these words may simply mean they read the treatment messages carefully. Therefore, besides showing that reading treatment messages would lead to increased support, what else should we conclude here? I am not saying the author is doing anything wrong here; I just think readers need more explanations and discussions to walk them through.

Editor:

On occasion we consult with the referees before we finalize our decision. In this case, we further consulted with Reviewer #1 regarding the technical critiques, especially regarding the sample, and the conceptual advance.

Reviewer #1:

I think the technical issues are with some merit. However, given this is a survey experiment, I don't feel that the technical issue is fatal. I think the author should address these issues in their manuscript if they are given the opportunity to revise and resubmit. I still think the manuscript has some merits on the conceptual advance.

Reviewer #2 (Remarks to the Author):

I am not very enthusiastic about this paper. As I see it, the data sources for the analysis are not representative, sometimes over-representing liberals, other times under-representing them, making comparisons across the various surveys difficult. Moreover, the basic findings are relatively weak. Finally, the paper has not done a very good job of understanding the mechanisms that undergird the findings from the experiment.

Let me begin with the surveys.

For Study 1, the sample is a convenience, opt-in sample comprised of highly diverse Asians but predominantly those of Chinese ancestry. The sample seems to over-represent the college educated (I don't understand why the author reports numbers of cases rather than percentages). The sample seems to also over-represent conservatives (and this matters — see the $-.32$ correlation in Table 1).

For Study 2, well educated and Chinese origin people were over-represented, just as young people, but the ideological bias is in the opposite direction, with liberals over-represented (and an even stronger relationship with conservatism in Table 2).

I also note quite different correlations with the dependent variable reported in Tables 1 and 2, with no explanations or discussion whatsoever.

I conclude that these opt-in samples clearly cannot support any inference to the larger population of

people of Asian ancestry.

With the different studies clearly varying in representativeness, I do not see much value to comparisons like those reported in Figure 3.

Four dependent variables are analyzed in Study 1. For two of the three preference variables, significant results are found; for the remaining indicator, the results are not significant. For the most important dependent variable (Black reparations), the results are just barely significant ($p = .033$). For the feasibility item, the results are not significant. Because two of the preference variables dominant the index, the results are significant. As shown in Figure 1 (but never discussed in terms of the actual breadth of support), these differences are all relatively small. I conclude that the experimental intervention had at best a weak and inconsistent effect on the dependent variables. And the strongest effect (general support for cash payments) could well have been due to respondents understanding the item to be asking about cash payments to those of Asian origin. For the two items connected to slavery, the p -values are $.033$ and $.135$. These are pretty weak findings.

Moreover, the weakness of the findings is at least in part owing to the fairly large percentage of respondents failing the manipulation check—as I understand it, about one-fourth of those in the intervention condition failed the check (page 12). This no doubt is a function of the very low quality of respondents in opt-in samples such as this one.

For Study 2, the results are a bit stronger, but no explanation is provided of the differences in the results of the two studies—except for ideology, with one study biased one way and the other biased the other way (page 21). Again, the results in Figure 2 suggest that none of these relationships is particularly strong.

Of course, this study, at best, represents the maximum possible effect of the intervention because all respondents in that condition were exposed to the argument. In reality, exposure to arguments like this is not commonplace, and those exposed are no doubt not randomly assigned to exposure.

In my view, the “qualitative analysis” (e.g., page 25) is essentially worthless, being as it is, based on a tiny number of respondents. The last full sentence on page 27 seems not to be supported by the analysis.

Nor can I see what specific evidence supports the “hypocrisy” hypothesis (as claimed on page 28).

So: where does this leave us?

I cannot support publication of this paper. The data sources are decidedly unrepresentative and in different ways. The findings are weak at best (on the slavery item in Figure 2, the difference across conditions is $.21$ on a measure that varies from 1 to 4). The paper demonstrates practically nothing about mechanisms. The external validity of the experiment is weak. All of these factors add up to the conclusion that the paper should not be published.

Reviewer #3 (Remarks to the Author):

This paper experimentally examined the role that reminders about reparations for Japanese internment played in increasing Asian Americans' support for redress for Black Americans. I think overall the question is important, the findings have the potential to be informative. However, I have a number of concerns about the theoretical background, literature review, and analyses. I discuss these below.

On page 4, lines 61 and 62, can the authors be more specific about what kinds of attitude change did these interventions lead to?

On page 4, lines 65 – 68, please use more descriptive terms (less jargon) or define the terms: common identity and sensemaking. Also, I think the word “is” is missing from the last clause.

In the studies described on page 4 lines 61 – 68, I am having trouble seeing that there are mixed results in these studies. Did common identity as a starting point lead to less or more support for reparation policy? How about nonjudgemental listening. I think more details in these studies are needed in order to support the primary point of this introduction.

On page 5, line 82, I am curious to learn more about Asian American support for Black reparations (since that is the focus of this study). Is 47% a high percentage? How does this number compare to White opposition?

On page 5, line 90, please tell us what “solution aversion” is without jargon, which would help the reader understand why this study is being discussed here and not in the above sections on page 4 with the other studies on interventions for policy support change. Also it seems like we need to understand “solution aversion” better since it is a possible theory behind the hypotheses of this study.

Similarly I am unclear why the studies on “moral hypocrisy” (please define) starting on page 5, line 96 are not included in the literature on page 4.

Overall I think the literature review needs some major reorganization. I am not clear if the studies described in the section on “Informational intervention and reminders of Japanese redress” are actually specifically related to Japanese redress (not enough study detail to discern) and similarly I am not sure if the studies described in the section on “Reparations payments for Black Americans” are specifically related to Black American redress. And if these studies are not (and I don't believe they are) I do not understand why the studies are organized into each of the sections the way they are.

Also, I think the lit review could use a broader scope on the history of Black and Asian relations. Much has been written about how these groups have been triangulated with White Americans and pitted in opposition.

Methods

Where were participants recruited from? Was this aided by MTurk or Prolific.co for example? Through social networking?

On page 10, line 223, provide the Cronbach alpha coefficient when item 4 is included so the reader can judge that removing this item is a reasonable thing to do.

I think that prior to your methods section on page 8, the authors should spell out the specific hypotheses in the main text of the paper and not just in the preregistration.

On the one hand, I appreciate the inclusion of all of the possible correlates and potential explanatory variables (e.g., conservatism, model minority work ethic, common fate, etc). However, without specific hypotheses it has a bit of a “kitchen sink” feel to it, where it seems the researchers have put a number of possibly related variables in a dataset but without any complex theoretical framing above that would help these relations make sense.

Typically a quantitative study would have a preliminary analysis section, exploring potential confounds that may need to be controlled. For example, how was the gender diversity within each experimental condition. If there is an imbalance across conditions, then gender would need to be controlled so it is not possible that gender is an explanation for findings. Similarly, how well is age balanced across experimental conditions? And, how is age related to support for Black reparations?

What is the purpose of study 2? Typically there is a logical progression from study 1 to study 2. Is there something the researchers missed in study 1 that could be helpful to explain the primary research question?

The text analysis is interesting to read, but not at all my area of expertise so I can not comment on the quality and robustness of it.

Dear Reviewers,

In this letter, we have responded to each of the comments raised by you. The structure of the letter is as follows: We start with a quotation from comments received in the decision letter from you and the reviewers. After each comment, we respond in bold and then provide quoted text, when relevant, from the manuscript that highlight our specific changes.

The primary changes to the manuscript include a significant overhaul of the introduction, new analyses involving what participants learned from the intervention to explore mechanisms, and acknowledgment throughout of the limits of our sampling procedures. We also expanded the authorship team to increase the speed of this revision.

The Authors

Reviewer #1 (Remarks to the Author):

#1: I appreciate the opportunity to review this article. In general, this article finds that reminding Asian Americans of the Japanese redress of Incarceration would increase Asian Americans' support for the redress of African American for slavery, and such effects are not through the change of political attitudes, perception of linked-fate, or anti-blackness.

I think this paper can make a great contribution to the study of Asian Americans as well as the study of public policy opinions. I think this paper has great potential and is well-written. I hope to make the following comments and suggestions.

Thank you for your kind words about the paper, and we agree that it could make an important contribution in the ways you describe.

#2: First, I would encourage the author to have more discussions on Asian American and their attitudes on inter-racial relations with African American and/or other minorities. As the author points out in p.29, there might be differences between Asian American and white Americans regarding the intervention effect. It makes sense given that, being a minority themselves, Asian Americans may have different opinions on how other minority groups (in this paper, African American) should be treated. Some recent studies and the Harvard Lawsuit seem to indicate that Asian Americans are not necessarily supportive of policies that benefit other minority groups, especially when they think those policies are benefiting other groups at the expense of Asian American interests. I believe engaging with the attitudes of Asian American toward other minority groups would be important for this paper.

This is an excellent suggestion. In the revision of the manuscript we more directly foreground the history of relations between Asian and Black Americans. You can find these changes beginning on page 3-4. Indeed, solidarity between Asian and Black Americans is not guaranteed, but exposure to reminders of redress is one pathway to greater solidarity:

“Understanding the history of relations between Asian and Black Americans is critical for a deeper understanding of movements for reparative justice. Though histories of racism differ considerably between these groups,

Asian and Black American intergroup relations have been characterized by solidarity. As early as 1955, groups representing Asian and African countries advocated for solidarity in charting a direction toward decolonization (Haddad-Fonda, 2017). In the U.S., Asian civil rights activists like Yuri Kochiyama and Thich Nhat Hanh worked in tandem with Black leaders like Malcom X and Martin Luther King Jr. to fight for racial justice (Fujino, 2005). Contemporary groups including Asians for Black Lives carry out this tradition of solidarity (Lang, 2020). Empirical studies of attitudes also bear out this solidarity: For instance, a majority of Asian Americans support the Black Lives Matter Movement (Goodman, Verma, & Phan, 2022).

Alongside this history of solidarity has been intergroup conflict. Scholars often discuss this conflict in terms of racial positioning, wherein two minoritized groups (e.g., Asian and Black Americans) are positioned against each other through rhetoric and policy in an attempt to drive discord between those groups (Kim, 1999; Poon et al., 2016; Zuo & Cheryan, 2017). For example, model minority stereotypes are one form of positioning, wherein Asian Americans are conceived of, through rhetoric and policy, as a “problem free” minority group, and then this stereotype is used to legitimize and maintain unjust racial hierarchy by suggesting that non-Asian minority groups themselves are the problem (e.g., Fields & Fields, 2012; Yoo, 2010). When these model minority messages are internalized by Asian Americans, conflict can arise on policies for achieving racial justice. For instance, in a 2023 survey by Pew, 52% of Asian Americans disapproved of selective colleges considering race in admissions decisions (i.e., affirmative action; Doherty, Kiley, Asheer, & Price, 2023). In contrast, 71% of Black Americans approved of the policy (Doherty et al., 2023). In this research, we attempt to better understand ways to promote solidarity and reduce ongoing potential conflict between Asian and Black Americans.”

#3: Second, I am a little unsure about the mechanism this paper tries to demonstrate. It seems acceptable to say that the treatment effects are not caused by any change in political attitudes or perceptions of African American, but what exactly is the mechanism? I feel it is a little weak to say the mechanism is “solution aversion” or “avoiding hypocrisy.” I would encourage more discussions and (if possible) more analysis on this.

The primary revisions to the paper have included discussion and analysis of mechanisms. Though we test several explanatory mechanisms in the original draft and this revision related to changes in political ideology, greater awareness of redress seems to statistically

mediate the relationship between the treatment and support for reparations. That is, the greater proportion of participants who now know about the amount of monetary redress granted to Japanese Americans that is increased by the intervention is related to support for redress, and mediates the relationship between the intervention and support for redress. You can find the analyses describing this mechanism on page 15-16 for Study 1 and an identical analysis in Study 2.

“To better understand whether learning about Japanese redress, versus some other shift in attitudes, explains greater support for reparations for Black Americans, we used Process Model 4 (Hayes, Montoya, & Rockwood, 2017), to conduct an exploratory mediation analysis with reparations support as the dependent variable, intervention as the independent variable, and responses to the quiz question about Japanese redress as the mediator. Because Process does not accept dichotomous mediators, quiz responses were rescored so that answering “yes, \$100,000” was rescored as “1,” a partially correct response, while “yes, \$20,000” was rescored as “2,” or a fully correct response, and “no” was rescored as “0,” or an incorrect response. In this analysis, we controlled for demographic variables related to education and gender. The latter variables were added to the model because education is confounded with knowledge, and gender was associated with reparations support in Study 2. Analyses without these controls were consistent with those reported here.

The model found a significant effect of the intervention on the quiz mediator $B = .605 (.101)$, $t(311) = 5.99$, $p < .001$, and a significant effect of the quiz mediator on reparations support $B = .109 (.049)$, $t(310) = 2.22$, $p = .0274$, as well as a significant effect of the intervention on reparations support in the same model $B = .195 (.093)$, $t(310) = 2.11$, $p = .0357$. Bootstrapping analysis with 5,000 resamples revealed a significant indirect effect of the intervention on reparations support through the acquire language mediator $B = .066(.0318)$ $CI_{95\%} [.0079, .1351]$. In the model, there was no credible evidence of relationships between education and reparations support, $B = -.0415 (.061)$, $t(310) = -0.680$, $p = .4973$, nor between gender and reparations support, $B = -.166 (.061)$, $t(310) = -1.887$, $p = .060$. Overall, we provide some statistical evidence consistent with our assertion that higher support for Black reparations in the intervention condition was statistically accounted for by increased knowledge of Japanese redress payments in that condition relative to the control condition.”

The introduction is now streamlined to highlight awareness as central to avoidance of hypocrisy, and you can find these revisions on page 6-7.

“A second goal of our research is to better understand why an intervention that provides knowledge of Japanese redress might increase Asian American support for Black reparations. We contend that learning about a history of redress for one’s own community will create conditions for moral hypocrisy that Asian Americans will seek to avoid by supporting similar redress for Black Americans.

Moral hypocrisy is derived from past social psychological research on cognitive dissonance (Aronson, Fried, & Stone, 1991). In that work, having participants (a) learn about or experience a positive outcome for their own group, and then (b) receive reminders of a failure of another group to experience that same or similar outcome induces a discrepancy called moral hypocrisy. Moral hypocrisy is an uncomfortable, dissonant state, and thus, people seek to resolve the discrepancy through changes in attitudes that create consistency in treatment between groups (Aronson et al., 1991; Bruneau et al., 2018). In recent research, this paradigm has been applied to religious groups: Participants were less likely to collectively blame all Muslims for transgressions of individual Muslims when reminded that they do not engage in similar collective blame for Christians (Bruneau, Kteily, & Falk, 2018). In follow-up research, a similar moral hypocrisy intervention reduced support for anti-Muslim policies (Gallardo et al., 2021).

We reason that providing information about the successful implementation of Japanese redress to a sample of Asian Americans and then asking them about unrealized redress for Black Americans due to chattel slavery and Jim Crow will engender a desire to avoid hypocrisy. We expect participants to avoid this discrepancy through an increase in support for reparations for Black Americans.”

#4: Third, I am not sure I fully understand Table 1 and the relevant discussions (also the same discussions in Study 2). Things like their level of conservatism and their network diversity should not be changed by the treatment material. Thus, if the experimental randomization is successful, we should see a balance across these items that are not changed by treatment. This means the level of conservatism and network diversity should be similar in the control and treatment groups. But if so, how can they be correlated with the support of reparation, which is clearly correlated with experimental groups?

A point of clarification is in order. Randomization in both studies would mean that the participants in treatment and control conditions are similar on several dimensions. We report this analysis as part of the revision on page 13. Indeed, there is good reason to believe that randomization was successful in our experiment based on this analysis (and an identical analysis performed in Study 2).

“To examine the success of our random assignment manipulation, we examined between group differences in demographic variables (i.e., education, income, gender, age, Japanese ancestry) that should not change as a function of our experimental manipulation. The result of this preliminary analysis indicates the success of our manipulation—there was no credible evidence for between condition differences in any of these demographic variables $t_s < 0.962$, $p_s > .337$.”

Important to our randomization, there is no statistically significant evidence that conservatism and network diversity are correlated with experimental group, but they are correlated with support for reparations. R1 may be referring to the sample differences in conservatism, wherein Study 1 did have a more conservative sample than Study 2? This happens because we used different online crowdsourcing platforms that sample in different ways and not because of a failure of random assignment.

#5: Fourth, the design of Study 2 can only show that Japanese redress itself can have some effects, but it cannot rule out that knowing the unfair treatment of Japanese also leads to increased support of Black reparations because no pure control groups are presented in experiment 2. Why is it designed in this way? Also, I think the author should be careful in claiming that only the Japanese reparation message works because we actually do not know whether the Japanese Incarceration messages also work or not.

A point of clarification is in order. The control condition in Study 2 is the first half of the treatment condition only, describing the unfair treatment of Japanese Americans during WWII. Thus, Study 2 provides a control condition wherein we know with greater confidence that it is the reparations piece of the treatment condition that increases support for reparations for Black Americans rather than just the unfair treatment part.

In terms of choices regarding the control group, we have tried to create control conditions that account for different alternative interpretations. In Study 1, the control group is exposed to a topic that is unrelated to the study issues, and as such, reparations support is likely to come from participants’ personal experiences and beliefs. The advantage of such a control condition is that any changes in the treatment are likely to be due to the treatment rather than the control condition. The problem with such a control condition, though, is that it leaves open alternative explanations that, for instance, one aspect of the treatment is more responsible for the effect on reparations support than another. We included the Study 2 control condition where we remind about unfairness of Japanese incarceration to determine if the redress information in particular shifted support for reparations. This interpretation is

consistent with our statistical analysis of results in Study 2. You can find clarification about the motivation for including the control conditions we have on page 18:

“In our second study, we asked a new sample of Asian Americans residing in the U.S. to take a 10-minute survey which randomly assigned each participant to our intervention, reminding participants of Japanese incarceration and redress, or to a control condition reminding participants about Japanese incarceration only. This control condition was chosen for Study 2 to rule out an alternative interpretation of our results in Study 1—that reminders about the injustice faced by Japanese Americans through their relocation and incarceration, and not the information about redress, changed participant attitudes toward Black reparations.”

and page 30:

“Importantly, we observed the success of our intervention informing about Japanese redress relative to an informational control about animals struggling with climate change and one that described Japanese incarceration (but not redress) in identical terms to the intervention. The latter finding with the incarceration only control condition suggests that it was the reminders of Japanese redress, in particular, that increased support for reparations for Black Americans.”

#6: Fifth, I don't feel the text analysis section is sufficiently convincing. It is natural to see more people write about getting justice in treatment groups in study 2 because that's what the treatment group's message is about. Similarly, for the quantitative analysis of texts, what can we conclude from the fact that “use of acquire related words were associated with support for reparations”? As said, the usage of these words may simply mean they read the treatment messages carefully. Therefore, besides showing that reading treatment messages would lead to increased support, what else should we conclude here? I am not saying the author is doing anything wrong here; I just think readers need more explanations and discussions to walk them through.

Our intention in including these analyses is to show that participants are now more aware of redress as a solution for past injustice. The text analysis is a way for us to examine this statistically, by asking participants about their reactions and impressions from the

experiment we can get a sense of how they interpret the intervention. The analysis is one way to do this. We think it is true that reading the treatment message, in particular, led to greater support, but it is particularly, the redress piece that is learned in the treatment condition. This is awareness that Asian Americans in our sample do not have until the treatment, and it is that awareness about past redress that we believe increase support for other reparative actions for Black Americans.

All this said, we agree that the discussion of these effects could be clearer and their meaning could be more precise, and that even with clearer writing these results are preliminary. In that vein, revisions to the paper have removed the qualitative and text analysis from the manuscript and placed them in the supplementary materials on page S2-S3. We think placing these analyses in the supplementary materials enhances the clarity of the manuscript.

Reviewer #2 (Remarks to the Author):

#1: I am not very enthusiastic about this paper. As I see it, the data sources for the analysis are not representative, sometimes over-representing liberals, other times under-representing them, making comparisons across the various surveys difficult.

We agree that a national probabilistic sample of Asian Americans would be a gold standard and superior to the samples we have collected here. In some of my lab's other survey research we have done this with related topics, but it is not something we field with every survey experiment because costs to field these national probability surveys are high (\$75k at NORC or Knowledge Networks and possibly more with the various language translations that might be necessary).

We view the surveys we have conducted here as essential preliminary work to these national probability surveys. For a fraction of the cost, we have conducted two interventions on different populations of Asian Americans. The two samples come from distinct online platforms that we have previous relationships with (Centiment and Prolific), and provide high quality data (e.g., Eyal et al., 2021). The results from the two studies are remarkably consistent given these demographic differences of the samples, which we view as a strength of the paper. That similar experimental results emerged with very different samples of Asian Americans increases our confidence in the generalizability of the results to a more representative sample of Asian Americans, and speaks to the importance of this kind of experimental prework.

Of course, we do think that future research and more careful sampling is ultimately necessary and you can find revisions that highlight this point on page 32.

#2: For Study 1, the sample is a convenience, opt-in sample comprised of highly diverse Asians but predominantly those of Chinese ancestry. The sample seems to over-represent the college educated (I don't understand why the author reports numbers of cases rather than percentages). The sample seems to also over-represent conservatives (and this matters — see the -.32 correlation in Table 1).

For Study 2, well educated and Chinese origin people were over-represented, just as young people, but the ideological bias is in the opposite direction, with liberals over-represented (and an even stronger relationship with conservatism in Table 2).

Thank you for drawing our attention to the differences between the samples. One of the strengths of our experiment, we believe, is that despite significant differences in conservatism between the samples our intervention has a similar effect on participant support for reparations, which suggests that the intervention can be successful in populations of Asian Americans that may be more or less conservative. We highlight this point on page 32 in our acknowledgment of a need for more research on this topic with representative and targeted samples.

#3: I also note quite different correlations with the dependent variable reported in Tables 1 and 2, with no explanations or discussion whatsoever.

Thank you for pointing out the importance of comparing the correlations across studies. When we examine the correlations in 7 of 10 correlations the direction (+ or -) of the correlation stays constant between Study 1 and 2. This is a reflection of the consistency in associations across our samples. Also, when comparing the magnitude of correlations using a Fisher R to Z transformation, we find evidence for only one statistically significant differences in magnitude of associations of the same correlation across the two studies, for conservatism, once we account for multiple unplanned comparisons using a Bonferroni correction ($\alpha = .005$). We report this analysis on page 27-28. These additional analyses suggest that there is no credible evidence for differences between correlations with reparations support (save for conservatism) across our studies.

“Moreover, exploratory comparisons of correlations using a Fisher r to z transformation with a Bonferroni correction ($\alpha = .005$) for multiple comparisons found no credible evidence for differences between conditions in correlations between support for reparations and gender, income, education, model minority beliefs, feelings toward Black or Asian Americans, common fate beliefs, or network diversity between the samples $z_s < 2.26$, $p_s > .0238$. One difference emerged in associations between conservatism between the two samples $z = 3.24$, $p = .0021$, with the Study 1 sample reporting higher mean conservatism, but lower associations between conservatism and support for reparations, which we explore below and in the supplementary analyses online.”

It is important to note, given R2s concern, our analysis on page 28, which shows that no statistical evidence for an interaction between study and condition emerged in our data $F(1,825) = 0.167$, $p = .682$.

#4: I conclude that these opt-in samples clearly cannot support any inference to the larger population of people of Asian ancestry.

We agree with R2 in that we think it is important to be careful to generalize our results to the larger population of Asian Americans. We note this important caution in the revision on page 32:

“This study was not without limitations that call for caution when interpreting these results beyond the online panels from which the studies were fielded. The Asian American category is a coalition of people from Asian origin countries with many unique immigration histories in the U.S. and languages. As such, our online panel samples have failed to fully capture the complexity of this broad collection of people. A national probability sample is warranted to better understand how the intervention we used here shapes attitudes toward reparations for Black Americans. As we mentioned above, targeted sampling is also warranted, to better understand how specific groups, such as Japanese Americans in particular, might respond more or less strongly to the intervention. Likewise, that the studies were conducted in English limits our conclusions to Asian Americans who would be likely to seek out opportunities to complete surveys in English.”

#5: With the different studies clearly varying in representativeness, I do not see much value to comparisons like those reported in Figure 3.

There are a few reasons why we continue to report the results as aggregated across our samples. Specifically, we planned to conduct the comparison across the studies in our pre-registered analysis, based on suggestions from prior research (Goh et al., 2016) and so deviating would be violating our analysis plan. Deviations from this plan could be permissible in some circumstances, but do not appear to be warranted here, because there is actually a lot of consistency in method, sampling, and results across the two studies. In terms of results, there is consistency between correlations across studies as the above analysis suggests. Moreover, there was also consistency in magnitude and direction of intervention effects on support for reparations. In terms of sampling, the studies are hampered by online crowdsourcing panels but do sample Asian Americans. In terms of methods, the experimental conditions are comparable across studies, critically, the treatment condition is identical. All these reasons make the pre-registered comparison across studies warranted and support our choice to conduct the combined analysis.

Figure 3 is also an important comparison for assessing how much our convenience samples are consistent, or not, with larger probability samples on support for reparations. As you can

see in the figure, our aggregate control condition shows no credible evidence for a difference in mean level of support for cash reparations in comparison to the UMass national probability sample asking the same question $t(421) = 0.717, p = .474, d = .035$ (page 29). If the control condition was significantly different from the UMass poll additional questions about representativeness would be warranted.

#6: Four dependent variables are analyzed in Study 1. For two of the three preference variables, significant results are found; for the remaining indicator, the results are not significant. For the most important dependent variable (Black reparations), the results are just barely significant ($p = .033$). For the feasibility item, the results are not significant. Because two of the preference variables dominant the index, the results are significant. As shown in Figure 1 (but never discussed in terms of the actual breadth of support), these differences are all relatively small. I conclude that the experimental intervention had at best a weak and inconsistent effect on the dependent variables. And the strongest effect (general support for cash payments) could well have been due to respondents understanding the item to be asking about cash payments to those of Asian origin. For the two items connected to slavery, the p-values are .033 and .135. These are pretty weak findings.

The effect of the composite was $d = .29$ in Study 1 and $d = .32$ in Study 2 and the direction of the treatment effect on each individual items that makes up the three-item composite was consistent across both studies. The notion that the intervention had a “weak and inconsistent effect” seems inconsistent with these data. Moreover, the effect size according to the d statistic is larger than the $d = .20$ that Cohen indicates as a small effect. Importantly small effects can be important effects: The association between taking aspirin and heart attack deaths is a correlation of $r = .02$, but when scaled to the nearly 800,000 Americans who die annually of heart attack, this small effect could literally mean life or death for thousands (see also Meyer et al., 2001).

For interpretation of effect size specifically in regard to support for cash payments to Black Americans we refer to Figure 3. In that figure we see that population support for reparations has increased from 15% in 2014 to 36% in 2021, which is an increase in 21 points over 7 years. Our intervention shows an increase in support for cash payments of nearly 10% across the two studies relative to the control condition, or an effect consistent with about three years of progressive attitude change on the topic. On page 29 we interpret the effect size and highlight some caution and promise in the magnitude of the effect of our intervention:

“Figure 3 also allows us to better understand and interpret the magnitude of the intervention’s impact on support for reparations for Black Americans. Though an effect size of $d = .313$ ($\eta_p^2 = .024$) is modest, effects need not be large to be important (e.g., Kraus, 2018; Meyer et al., 2001). If we compare the difference in reparations support between the control and intervention across our studies we see that there is nearly a 10% difference in support for reparations payments, which is roughly equivalent to 46.7% of the total progressive attitude change in reparations support from 2014 to 2021.”

Only the last item about Black reparations occurring in “my lifetime” showed inconsistent results across studies—but this item did not covary with the others and we articulate reasons why (articulated on page 31), whether reparations for Black Americans will happen in my lifetime, might be shaped differentially by the intervention:

“Interestingly, reminders of the Japanese struggle for redress did not reliably change people’s conceptions of whether Black reparations were less, versus more likely in “my lifetime,” relative to the control condition. Prior research indicates that Americans tend to be more optimistic about positive change toward racial equality than data trends would suggest (DeBell, 2017; Kraus et al., 2017), but perhaps that tendency is countered by being confronted with the political realities of the 45-year struggle for Japanese redress. More research is necessary to better understand theories of progress and change among Asian Americans and in the context of reparations specifically.”

#7: Moreover, the weakness of the findings is at least in part owing to the fairly large percentage of respondents failing the manipulation check—as I understand it, about one-fourth of those in the intervention condition failed the check (page 12). This no doubt is a function of the very low quality of respondents in opt-in samples such as this one.

It is important to carefully examine the quality of data and we have scrutinized our data in the current study in several ways. For instance, despite the fact that the quiz question was asked toward the end of the survey amongst other items and that the intervention included many different pieces of information about Japanese incarceration, we did see significantly greater accuracy in the intervention than in the control condition across both studies, and an impressive 91.9% correct response rate in the intervention condition in Study 2. Moreover, if we recode correct responses to indicate partial correct, for instance, saying “yes, Japanese redress did occur”, the percentage of correct responses increases to 80.4% for Study 1 and 93.5% for Study 2. We find this to be compelling evidence that participants were attentive to the survey, and that many learned about Japanese redress through the manipulation. You can see this analysis on page 14 and 22.

The rich qualitative data from Study 2 is also indicative of careful responding (now reported in the supplementary materials). That several people were connecting Japanese redress to Black reparations reflects how much participants were engaging with our materials. Black Americans were not explicitly mentioned in the intervention.

#8: For Study 2, the results are a bit stronger, but no explanation is provided of the differences in the results of the two studies—except for ideology, with one study biased one way and the other

biased the other way (page 21). Again, the results in Figure 2 suggest that none of these relationships is particularly strong.

We thought it was important to test if there were condition differences between studies in the original manuscript and revision. It is important to note our analysis on page 28, which shows that no statistical evidence for an interaction between study and condition emerged in our analysis $F(1,825) = 0.167, p = .682$. In terms of the relationships not being particularly strong, please see our response to the prior points about small effects and their importance (#6).

#9: Of course, this study, at best, represents the maximum possible effect of the intervention because all respondents in that condition were exposed to the argument. In reality, exposure to arguments like this is not commonplace, and those exposed are no doubt not randomly assigned to exposure.

Data from our paper suggests that Study 2 results were not unusually large or an upper bound for this kind of intervention. The effect of the composite reparations item was $d = .29$ in Study 1 and $d = .32$ in Study 2 and the direction of the treatment effect on each individual item that makes up the three-item composite was consistent across both studies. Moreover, as per our original analysis no statistical evidence for an interaction between study and condition emerged $F(1,825) = 0.167, p = .682$ (page 29). In some of our prior work we have conducted these interventions in laboratory settings (Callaghan et al., 2021), and those in-person interventions tend to produce larger effects on attitudes because of the increased immersion possible in a laboratory. This kind of intervention would be difficult to conduct in person, though, because sampling Asian Americans in our laboratory sample would require a significant additional community sample outreach project. We note the potential for in-person targeted sampling on page 32.

R2 argues that “exposure to arguments like this is not commonplace.” This is central to our study rationale and something we argue in the introduction—that many Japanese Americans did not discuss redress. Many Japanese American families, include the family of the first author (MK) of this manuscript, did not talk about their experiences around the threat of incarceration (see Murray, 2002). The point about the intervention not being commonplace is why conducting this research is necessary, to determine if such messaging, if it were made more common in schools and history classes, might increase support for reparations. We note this on page 6:

“Evidence indicates that there is a lack of knowledge of Japanese redress among Americans, and Asian Americans in particular, which indicates that the informational intervention may be effective: A YouGov 2014 poll found that a minority of Americans (37%) were in favor of Japanese reparations payments, a finding which reflects that many Americans may not know about the history of these payments already having occurred (Moore, 2014), or much Asian American US history in general (Lee, 2022). This is compounded by the fact that many Japanese people who were

incarcerated were unwilling to talk about this period of their lives within their own communities (Miki, 2004; Murray, 2022)."

#10: In my view, the "qualitative analysis" (e.g., page 25) is essentially worthless, being as it is, based on a tiny number of respondents. The last full sentence on page 27 seems not to be supported by the analysis.

Given the clear questions about the worth of the qualitative and text analysis we report in the paper, our inclination is to move these analyses to the supplementary materials on page S3. Moving these analyses online improves the clarity of the paper.

We still find these analyses to be useful. The use of our qualitative analysis is exploratory and preliminary and can give us an idea of how participants are processing our intervention. Scholars do differ in how they interpret qualitative evidence. We find it informative to see that what people learn about Japanese redress in survey responses is also consistent with how they write about the intervention at the end of the survey in open-ended items. We clarify the purpose of these qualitative and LIWC analyses on page S3.

The last sentence on page 27 no longer appears in the manuscript.

#11: Nor can I see what specific evidence supports the "hypocrisy" hypothesis (as claimed on page 28).

The logic for hypocrisy driving these effects is that the knowledge that one group who has received redress for past injustice should drive support for redress for another group to avoid hypocrisy. We have clarified this point on page 6-7:

"Moral hypocrisy is derived from past social psychological research on cognitive dissonance (Aronson, Fried, & Stone, 1991). In that work, having participants (a) learn about or experience a positive outcome for their own group, and then (b) receive reminders of a failure of another group to experience that same or similar outcome induces a discrepancy called moral hypocrisy. Moral hypocrisy is an uncomfortable, dissonant state, and thus, people seek to resolve the discrepancy through changes in attitudes that create consistency in treatment between groups (Aronson et al., 1991; Bruneau et al., 2018). In recent research, this paradigm has been applied to religious groups: Participants were less likely to collectively blame all Muslims for transgressions of individual Muslims when reminded that they do not engage in similar collective blame for Christians (Bruneau, Kteily, & Falk, 2018). In follow-up research, a similar moral hypocrisy intervention reduced support for anti-Muslim policies (Gallardo et al., 2021).

We reason that providing information about the successful implementation of Japanese redress to a sample of Asian Americans and then asking them about unrealized redress for Black Americans due to chattel slavery and Jim Crow will engender a desire to avoid hypocrisy. We expect participants to avoid this discrepancy through an increase in support for reparations for Black Americans.”

Reviewer #3 (Remarks to the Author):

#1: This paper experimentally examined the role that reminders about reparations for Japanese internment played in increasing Asian Americans’ support for redress for Black Americans. I think overall the question is important, the findings have the potential to be informative. However, I have a number of concerns about the theoretical background, literature review, and analyses. I discuss these below.

On page 4, lines 61 and 62, can the authors be more specific about what kinds of attitude change did these interventions lead to?

We specify that we are interested in studying attitude changes related to support for Black reparations throughout the introduction. See changes on page 3:

“In this research, we ask whether knowledge of Japanese redress would increase Asian American support for another movement for reparative economic justice: Reparations payments for Black Americans for chattel slavery and Jim Crow (Darity & Mullen, 2020).”

#2: On page 4, lines 65 – 68, please use more descriptive terms (less jargon) or define the terms: common identity and sensemaking. Also, I think the word “is” is missing from the last clause.

We add explanation and simplify terminology throughout the introduction in the revision on page 8.

“The study also helps clarify how informational interventions might shape policy attitudes. Interventions intended to teach an accurate history of American racism have mixed evidence with respect to their effectiveness in policy discussions (e.g., Hetey & Eberhardt, 2018). Past research indicates that an intervention that provides new information, and also takes steps to message in ways that reduce defensiveness can be effective in changing attitudes (Broockman & Kalla, 2016; Callaghan et al., 2021). Our intervention is informed by this past research and takes

several steps, discussed later, to reduce defensiveness.”

#3: In the studies described on page 4 lines 61 – 68, I am having trouble seeing that there are mixed results in these studies. Did common identity as a starting point lead to less or more support for reparation policy? How about nonjudgemental listening. I think more details in these studies are needed in order to support the primary point of this introduction.

We have largely removed the section on nonjudgemental listening and sensemaking from the revision. You can find a brief reference to these studies on page 8 (reproduced above).

#4: On page 5, line 82, I am curious to learn more about Asian American support for Black reparations (since that is the focus of this study). Is 47% a high percentage? How does this number compare to White opposition?

We have added a new section on history of solidarity between Asian and Black Americans with information on Asian American support for Black reparations on page 3-4, and reproduced above in R1s comments.

#5: On page 5, line 90, please tell us what “solution aversion” is without jargon, which would help the reader understand why this study is being discussed here and not in the above sections on page 4 with the other studies on interventions for policy support change. Also it seems like we need to understand “solution aversion” better since it is a possible theory behind the hypotheses of this study.

We have removed the section on solution aversion to more appropriately contextualize the results in terms of moral hypocrisy avoidance the arises from learning information about Japanese redress. The introduction is completely reorganized but you can find these changes on page 6.

#6: Similarly I am unclear why the studies on “moral hypocrisy” (please define) starting on page 5, line 96 are not included in the literature on page 4.

We have reorganized the introduction so that the moral hypocrisy section follows from the the prior section about knowledge on page 6-7. We keep the sections separately so that we have sufficient space to discuss why knowledge and moral hypocrisy relate to each other:

“A second goal of our research is to better understand why an intervention that provides knowledge of Japanese redress might increase Asian American support for Black reparations. We contend that learning about a

history of redress for one's own community will create conditions for moral hypocrisy that Asian Americans will seek to avoid by supporting similar redress for Black Americans.

Moral hypocrisy is derived from past social psychological research on cognitive dissonance (Aronson, Fried, & Stone, 1991). In that work, having participants (a) learn about or experience a positive outcome for their own group, and then (b) receive reminders of a failure of another group to experience that same or similar outcome induces a discrepancy called moral hypocrisy. Moral hypocrisy is an uncomfortable, dissonant state, and thus, people seek to resolve the discrepancy through changes in attitudes that create consistency in treatment between groups (Aronson et al., 1991; Bruneau et al., 2018). In recent research, this paradigm has been applied to religious groups: Participants were less likely to collectively blame all Muslims for transgressions of individual Muslims when reminded that they do not engage in similar collective blame for Christians (Bruneau, Kteily, & Falk, 2018). In follow-up research, a similar moral hypocrisy intervention reduced support for anti-Muslim policies (Gallardo et al., 2021).

We reason that providing information about the successful implementation of Japanese redress to a sample of Asian Americans and then asking them about unrealized redress for Black Americans due to chattel slavery and Jim Crow will engender a desire to avoid hypocrisy. We expect participants to avoid this discrepancy through an increase in support for reparations for Black Americans.”

#7: Overall I think the literature review needs some major reorganization. I am not clear if the studies described in the section on “Informational intervention and reminders of Japanese redress” are actually specifically related to Japanese redress (not enough study detail to discern) and similarly I am not sure if the studies described in the section on “Reparations payments for Black Americans” are specifically related to Black American redress. And if these studies are not (and I don’t believe they are) I do not understand why the studies are organized into each of the sections the way they are.

We have reorganized the literature review around the history of Asian and Black American solidarity, and the promise of an informational intervention for increasing support for Black reparations. These changes begin on page 3.

#8: Also, I think the lit review could use a broader scope on the history of Black and Asian relations. Much has been written about how these groups have been triangulated with White Americans and pitted in opposition.

The history section on Black and Asian American relations starts on page 3, and broadens our literature review.

#9: Methods

Where were participants recruited from? Was this aided by MTurk or Prolific.co for example? Through social networking?

In the original manuscript and revision we provide more detail about the online crowdsourcing samples used from Centiment surveys and Prolific. The revisions occur on page 9 and page 18-19:

“We also used a distinct, sign-up based online crowdsourcing survey panel, with a reputation for high data quality (Douglas et al., 2023; Eyal et al., 2021), to recruit Asian American participants for Study 2, to examine if our findings may generalize to this new group of Asian American participants.”

#10: On page 10, line 223, provide the Cronbach alpha coefficient when item 4 is included so the reader can judge that removing this item is a reasonable thing to do.

The alpha for all four items is .56 in Study 1 and rises to .82 when deleting the last item. We include these alpha levels in the revision on page 12.

#11: I think that prior to your methods section on page 8, the authors should spell out the specific hypotheses in the main text of the paper and not just in the preregistration.

The revision includes a specific statement about the hypotheses on page 7:

“In the current study, we test the central hypothesis that exposure to information about redress for Japanese incarceration, versus a control condition, would elicit increased support for reparations for Black Americans. We test this central prediction in two preregistered studies.”

#12: On the one hand, I appreciate the inclusion of all of the possible correlates and potential explanatory variables (e.g., conservatism, model minority work ethic, common fate, etc). However, without specific hypotheses it has a bit of a “kitchen sink” feel to it, where it seems the researchers have put a number of possibly related variables in a dataset but without any complex theoretical framing above that would help these relations make sense.

In the revision we have briefly discussed how the intervention might shape these political attitudes, and we continue to include these analyses to demonstrate the discriminant validity of our results. The revision is included on page 8.

“In addition to our attempts to measure how reminders of Japanese redress shape attitudes toward Black reparative economic justice, we also measured several constructs that might explain why the intervention changed Asian American attitudes. In particular, we assessed whether reminders of Japanese redress, because they highlight shared experiences between Asian Americans and other marginalized populations, increase feelings of common fate between Asian and Black Americans (e.g., the success of Asian people depends on the success of Black people), reduce anti-black attitudes, increase perceptions of network diversity, reduce conservatism, and reduce internalization of model minority stereotypes (e.g., Asian Americans work harder than other groups). Across each of these constructs, we explored whether the intervention shifted endorsement of these attitudes, as well as, whether these attitudes moderated the impact of the intervention (e.g., Jefferson & Ray, 2022; Yoo et al., 2010).”

#13: Typically a quantitative study would have a preliminary analysis section, exploring potential confounds that may need to be controlled. For example, how was the gender diversity within each experimental condition. If there is an imbalance across conditions, then gender would need to be controlled so it is not possible that gender is an explanation for findings. Similarly, how well is age balanced across experimental conditions? And, how is age related to support for Black reparations?

We have included a preliminary analysis section to highlight any imbalance in experimental conditions, which we did not observe in the study. You can find this revision on page 13-14 (Study 1) and page 21-22 (Study 2).

“To examine the success of our random assignment manipulation, we examined between group differences in demographic variables (i.e., education, income, gender, age, Japanese ancestry) that should not change as a function of our experimental manipulation. The result of this preliminary analysis indicates the success of our manipulation—there was no credible evidence for between condition differences in any of these demographic variables $t_s < 0.962$, $p_s > .337$.

If our intervention is likely to shape attitudes toward reparations, a preliminary step is that it must first inform Asian American participants about the existence of Japanese redress payments. We expected and found that

participants answered the quiz question correctly more in the intervention condition (74.1%) than the control condition (45.9%), $X^2(1) = 26.95, p < .001$, indicating that our manipulation was successful in informing participants about Japanese redress. Moreover, when we recoded quiz responses to give correct scores to responses that also included answers where the amount paid was incorrect, but answered that the redress payments had occurred, the proportion of correct responses in the intervention condition rose further to 80.4%.”

#14: What is the purpose of study 2? Typically there is a logical progression from study 1 to study 2. Is there something the researchers missed in study 1 that could be helpful to explain the primary research question?

In the revision we have included a brief statement outlining the changes to the control condition in Study 2, and the new online crowdsourced sample, that answer specific questions raised in Study 1. These revisions occur on page 18:

“In our second study, we asked a new sample of Asian Americans residing in the U.S. to take a 10-minute survey which randomly assigned each participant to our intervention, reminding participants of Japanese incarceration and redress, or to a control condition reminding participants about Japanese incarceration only. This control condition was chosen for Study 2 to rule out an alternative interpretation of our results in Study 1—that reminders about the injustice faced by Japanese Americans through their relocation and incarceration, and not the information about redress, changed participant attitudes toward Black reparations.”

#15: The text analysis is interesting to read, but not at all my area of expertise so I can not comment on the quality and robustness of it.

In the revision we try to interpret the qualitative and text analysis with greater caution, and interpret the preliminary findings as indicative of what participants learned from the intervention. These changes appear in the supplementary materials on page S3-S4.

14th Sep 23

Dear Professor Kraus,

Thank you for your patience during the peer-review process. Your manuscript titled "Reminders of Japanese redress and Asian American support for Black reparations" has now been seen by 2 of the original reviewers, Reviewer #1 and Reviewer #3. Reviewer #2 declined the invitation to re-review. I include the comments we received on your revision at the end of this message. They find your revised manuscript much improved, but a few important points remain. We are interested in the possibility of publishing your study in *Communications Psychology*, but would like to consider your responses to these concerns and assess a revised manuscript before we make a final decision on publication.

We therefore invite you to revise and resubmit your manuscript, along with a point-by-point response to the reviewers. Please highlight all changes in the manuscript text file.

Editorially, we consider it important that you address the remaining concerns from Reviewer #3. If you had a priori hypotheses, regardless of whether these were confirmed, please explicitly state these. Otherwise, please clearly lay out your research questions such that the reader can clearly see the link between the research questions and the analyses completed, highlighting that you had no a priori formulated hypotheses.

When reporting null results please use appropriate language to describe the results. (There is no statistical test that can demonstrate absence of an effect. Statements such as 'There is no difference between x and y.' or 'X does not affect Y.' must be revised to read 'We found [no/little] credible evidence of a difference between x and y.' or 'We found [no/little] credible evidence that X affects Y.')

Editorial Policy: [Policy requirements](https://www.nature.com/documents/nr-editorial-policy-checklist.pdf) (Download the link to your computer as a PDF.)

Please note that your revised manuscript must comply with our formatting and reporting requirements, which are summarized on the following checklist: [Communications Psychology formatting checklist](https://www.nature.com/documents/commspsychol-style-formatting-checklist-article-rr.pdf) and also in our style and formatting guide [Communications Psychology formatting guide](https://www.nature.com/documents/commspsychol-style-formatting-guide-accept.pdf) .

Please use the following link to submit your revised manuscript, point-by-point response to the

referees' comments (which should be in a separate document to any cover letter) and the completed checklist:
[link redacted]

Please do not hesitate to contact me if you have any questions or would like to discuss these revisions further. We look forward to seeing the revised manuscript and thank you for the opportunity to review your work.

Best regards,

Jennifer Bellingtier

Jennifer Bellingtier, PhD
Senior Editor
Communications Psychology

EDITORIAL POLICIES AND FORMATTING

* **CODE AVAILABILITY:** All Communications Psychology manuscripts must include a section titled "Code Availability" at the end of the methods section. In the event of publication, we require that the custom analysis code supporting your conclusions is made available in a publicly accessible repository; at publication, we ask you to choose a repository that provides a DOI for the code; the link to the repository and the DOI will need to be included in the Code Availability statement. Publication as Supplementary Information will not suffice. We ask you to prepare code at this stage, to avoid delays later on in the process.

*** DATA AVAILABILITY:**

All Communications Psychology manuscripts must include a section titled "Data Availability" at the end of the Methods section or main text (if no Methods). More information on this policy, is available at <http://www.nature.com/authors/policies/data/data-availability-statements-data-citations.pdf>.

At a minimum the Data availability statement must explain how the data can be obtained and whether there are any restrictions on data sharing. Communications Psychology strongly endorses open sharing of data. If you do make your data openly available, please include in the statement:

We recommend submitting the data to discipline-specific, community-recognized repositories, where possible and a list of recommended repositories is provided at <http://www.nature.com/sdata/policies/repositories>.

If a community resource is unavailable, data can be submitted to generalist repositories such as [figshare](https://figshare.com/) or [Dryad Digital Repository](http://datadryad.org/). Please provide a unique identifier for the data (for example a DOI or a permanent URL) in the data availability statement, if possible. If the repository does not provide identifiers, we encourage authors to supply the search terms that will return the data. For data that have been obtained from publicly available sources, please provide a URL and the specific data product name in the data availability statement. Data with a DOI should be further cited in the methods reference section.

REVIEWERS' EXPERTISE:

Reviewer #1 Intergroup Relations

Reviewer #3 Minority Groups, Intergroup Relations

REVIEWERS' COMMENTS:

Reviewer #1 (Remarks to the Author):

I have no further issues. Appreciate the efforts the authors made to revise the manuscript

Reviewer #3 (Remarks to the Author):

This is a much improved paper and I think can make an important contribution to the field. I do have some remaining questions.

- I still don't see formally laid out hypotheses. I think this would be the editor's choice given that whether or not to include formal hypotheses seems to depend on the preference of the journal.

- In study 1, I still don't see preliminary analyses examining potential confounding demographics. I do see that the authors state on page 16 that gender and education were related to knowledge as determined in study 2. There are a couple of reasons this is confusing to me. First, does this mean study 2 was completed before study 1? If so this should be clarified and explained why they are presented in the order that they are. Also I see that on page 22, the authors' state there was no significant difference by demographic variables? Does this contradict what was said on page 16? Please clarify.

- I am still feeling a bit that this is a "kitchen sink" analysis, which is why I think formal hypotheses would be valuable. For example, why are responses to the attention quiz tested as a mediator but not the other extraneous variables such as common fate and conservatism? Would it make more sense to analyze all of these potential mediators in a parallel mediator analysis?

Dear Reviewers,

In this letter, we have responded to each of the comments raised by you. The structure of the letter is as follows: We start with comments received in the decision letter from the reviewers in bold. After each comment, we respond and then provide quoted text in italics, when relevant, from the manuscript that highlights our specific changes.

The Authors

Reviewer #3 (Remarks to the Author)

#1 State the formal hypotheses:

Apologies! We did not fully understand what you meant by explicitly stating the formal hypotheses and this may have been due to disciplinary differences or some other misreading of your last comments. As you will see, the revised manuscript makes it clear that we test two Hypotheses. The first is our formally preregistered prediction:

The above analysis sets the stage for our central hypothesis. We tested this hypothesis in two preregistered studies:

Hypothesis I: Exposure to information about redress for Japanese incarceration, versus a control condition, will elicit increased support for reparations for Black Americans.

We also have a second exploratory hypothesis that was generated in revisions based on reviewer feedback which is also explicitly stated at the end of the introduction:

Learning about Japanese redress is critical for creating the conditions for moral hypocrisy, which we expect to be necessary for changes in support for Black reparations. Thus, we also explored a second hypothesis, which we tested by examining responses to a quiz about Japanese redress administered at the end of each study:

Hypothesis II: The extent of learning about Japanese redress in the intervention condition will account for the intervention's impact on support for Black reparations.

#2 I didn't see preliminary analyses to the experiments:

You can find the preliminary analyses at the very start of both results sections in Study 1 and 2. For Study 1, we report the results below:

To examine the success of our random assignment manipulation, we examined between group differences in demographic variables (i.e., education, income, gender, age, Japanese ancestry) that should not change as a function of our experimental manipulation. The result of this preliminary analysis indicates the success of our manipulation—there was no credible evidence for between condition differences in any of these demographic variables $t_s < 0.962$, $p_s > .337$.

In Study 2 we report the results below:

To examine the success of our random assignment manipulation, we examined between group differences in demographic variables (i.e., education, income, gender, age, Japanese ancestry) that should not change as a function of our experimental manipulation. The result of this preliminary analysis indicates the success of our manipulation—there was no credible evidence for between condition differences in any of these demographic variables $t_s < 1.051$, $p_s > .147$.

#3 Why are some variables tested for mediation but not others:

The mediation analysis explores what other variables, that shifted as a function of the intervention, may impact reparations support. As we report, our political attitudes measures do not shift as a function of the intervention, so the results of a mediation with those variables are likely to be equivocal with that analysis. That is, if there is no evidence for an intervention changing a political attitude, it is unlikely that the attitude impacts reparations support as a function of the intervention. Of course, these political attitudes can and do relate to reparations support, a fact we report in the political attitude correlations at the end of each results section. Based on this reasoning, we did not conduct or report additional mediation analyses.

The reviewer suggests testing multiple mediators in the same model, but this creates significant multicollinearity issues, particularly as the political attitude measures are highly correlated with the reparations support measure (Blalock Jr., 1965). Thus, multiple mediator models with political attitudes would be difficult to interpret and as such we have chosen not to include those in our revision.

#4 What was the order of the studies?

Study 1 was conducted before Study 2, but we did perform re-analyses of the data after both studies were completed so that our insights from Study 2 and reviewer comments could inform our analysis of Study 1. This is why we refer to Study 2 analyses in Study 1. I hope this answers your question.

#5 Was there a contradiction between what is said on p. 16 and p. 22?

There is no contradiction, condition differences on demographics do not emerge in either study. Gender is associated with reparations support (the DV) in Study 2 and not the intervention. We hope this answers your question.

28th Sep 23

Dear Professor Kraus,

Your manuscript titled "Reminders of Japanese redress and Asian American support for Black reparations" has now been seen by Reviewer 3, whose comments appear below. In light of their advice I am delighted to say that we are happy, in principle, to publish a suitably revised version in *Communications Psychology* under the open access CC BY license (Creative Commons Attribution v4.0 International License).

We therefore invite you to revise your paper one last time to address a list of editorial requests. At the same time we ask that you edit your manuscript to comply with our format requirements and to maximise the accessibility and therefore the impact of your work.

EDITORIAL REQUESTS:

SUBMISSION INFORMATION:

OPEN ACCESS:

Communications Psychology is a fully open access journal. Articles are made freely accessible on publication under a [CC BY license](http://creativecommons.org/licenses/by/4.0) (Creative Commons Attribution 4.0 International License). This license allows maximum dissemination and re-use of open access materials and is preferred by many research funding bodies.

For further information about article processing charges, open access funding, and advice and support from Nature Research, please visit <https://www.nature.com/commspsychol/article-processing-charges>

At acceptance, you will be provided with instructions for completing this CC BY license on behalf of all authors. This grants us the necessary permissions to publish your paper. Additionally, you will be asked to declare that all required third party permissions have been obtained, and to provide billing information in order to pay the article-processing charge (APC).

* **DATA AVAILABILITY:**

[link redacted]

Best regards,

Jennifer Bellingtier

Jennifer Bellingtier, PhD
Senior Editor
Communications Psychology

REVIEWERS' EXPERTISE:

Reviewer #3 Minority Groups, Intergroup Relations

REVIEWERS' COMMENTS:

Reviewer #3 (Remarks to the Author):

I am satisfied with the authors' edits and would support publication of this article